# ClpP protease activation results from the reorganization of the electrostatic interaction networks at the entrance pores

Mark F. Mabanglo[1,10], Elisa Leung[1,10], Siavash Vahidi[1,2,3,4,10], Thiago V. Seraphim[1,5,10], Bryan T. Eger[1], Steve Bryson[1,6], Vaibhav Bhandari[1], Jin Lin Zhou[3], Yu-Qian Mao[1], Kamran Rizzolo[1], Marim M. Barghash[1], Jordan D. Goodreid[3], Sadhna Phanse[1,5], Mohan Babu[5], Leandro R.S. Barbosa [7], Carlos H.I. Ramos[8], Robert A. Batey[3], Lewis E. Kay[1,2,3,4], Emil F. Pai[1,2,6,9] & Walid A. Houry [1,3]*

Bacterial ClpP is a highly conserved, cylindrical, self-compartmentalizing serine protease required for maintaining cellular proteostasis. Small molecule acyldepsipeptides (ADEPs) and activators of self-compartmentalized proteases 1 (ACP1s) cause dysregulation and activation of ClpP, leading to bacterial cell death, highlighting their potential use as novel antibiotics. Structural changes in *Neisseria meningitidis* and *Escherichia coli* ClpP upon binding to novel ACP1 and ADEP analogs were probed by X-ray crystallography, methyl-TROSY NMR, and small angle X-ray scattering. ACP1 and ADEP induce distinct conformational changes in the ClpP structure. However, reorganization of electrostatic interaction networks at the ClpP entrance pores is necessary and sufficient for activation. Further activation is achieved by formation of ordered N-terminal axial loops and reduction in the structural heterogeneity of the ClpP cylinder. Activating mutations recapitulate the structural effects of small molecule activator binding. Our data, together with previous findings, provide a structural basis for a unified mechanism of compound-based ClpP activation.

[1] Department of Biochemistry, University of Toronto, Toronto, Ontario M5G 1M1, Canada. [2] Department of Molecular Genetics, University of Toronto, Toronto, Ontario M5S 1A8, Canada. [3] Department of Chemistry, University of Toronto, Toronto, Ontario M5S 3H6, Canada. [4] Program in Molecular Medicine, The Hospital for Sick Children Research Institute, Toronto, Ontario M5G 0A4, Canada. [5] Department of Biochemistry, University of Regina, Regina, Saskatchewan S4S 0A2, Canada. [6] Ontario Cancer Institute/Princess Margaret Hospital, Campbell Family Institute for Cancer Research, Toronto, Ontario M5G 1L7, Canada. [7] Institute of Physics, University of São Paulo, São Paulo SP 05508-090, Brazil. [8] Institute of Chemistry, University of Campinas UNICAMP, Campinas SP 13083-970, Brazil. [9] Department of Medical Biophysics, University of Toronto, Toronto, Ontario M5S 1A8, Canada. [10] These authors contributed equally: Mark F. Mabanglo, Elisa Leung, Siavash Vahidi, Thiago V. Seraphim. *email: walid.houry@utoronto.ca

The caseinolytic protease P (ClpP) is a tetradecameric serine protease comprised of two heptameric rings stacked together in a cylindrical structure. It is one of the main proteolytic complexes in bacteria[1]. It associates with hexameric unfoldase chaperones of the AAA+ (ATPases Associated with diverse cellular Activities) superfamily such as ClpX and ClpA in *Escherichia coli* that selectively bind, unfold, and translocate target proteins through the axial pores of ClpP and into the proteolytic chamber for degradation[2]. The proteolytic fragments are then expelled through transient side pores or the axial pores[3–6]. In the absence of the ATPases, ClpP can only degrade unstructured or poorly folded proteins and small peptides of up to about 30 residues[7].

Structures of wild-type (WT) ClpP from different organisms as well as ClpP with mutations, substrate mimics, and small molecules have been determined[3,6,8–22]. Across species, ClpP has a conserved architecture composed of an N-terminal axial loop, a core head domain containing a Ser-His-Asp catalytic triad, and an equatorial handle region mediating tetradecamer assembly by interdigitation. ClpP has been crystallized in three distinct conformations: extended, compact, and compressed, highlighting its plasticity[23]. The main difference between these states involves the twisting and compaction of the ClpP barrel, facilitated by structural changes in the handle region found at the ring–ring interface[6,23].

Small-molecule activators of ClpP have been identified and have shown activities against several bacterial species and cancer cell lines, suggesting their exciting potential as antimicrobial and antineoplastic drugs[15,17,19,24–39]. Brötz-Oesterhelt et al.[24] discovered acyldepsipeptides (ADEPs) that activate ClpP in the absence of AAA+ chaperones, causing unspecific degradation of cellular proteins (i.e. dysregulation of ClpP) and ultimately cell death. The ADEPs bind with high affinity to hydrophobic pockets on the surface of ClpP, analogous to the binding of ClpX's IGF loops[15]. While, initially, ADEPs exhibited limited antibacterial activities[24], optimization of their chemical structure has led to analogs with improved potency against Gram-positive bacteria. In Gram-negative species, active efflux and limited penetration of the outer membrane caused low potencies[24]. Further optimization of ADEPs by our group led to a new derivative with improved activities against Gram-positive species, *S. aureus* and *E. faecalis*, as well as potency against select Gram-negative species, *Neisseria meningitidis* and *Neisseria gonorrhoeae*[19]. Moreover, recent efforts in our laboratory to develop other ClpP dysregulators targeting Gram-negative bacteria have led to the discovery of a new class of potential antibiotics named as activators of self-compartmentalizing proteases 1 (ACP1s)[28]. Like ADEPs, ACP1s bind to ClpP and exhibit comparable bactericidal properties as ADEPs but are easier to synthesize than ADEPs.

Here, we report, to the best of our knowledge, the first crystal structure of an ACP1 activator with EcClpP, showing binding modes that can be used to guide future drug optimization studies. We also report the crystal structure of *N. meningitidis* ClpP (NmClpP) with a novel ADEP analog, as well as of activated NmClpP mutants that mimic the structural and kinetic properties of ADEP- and ACP1-activated ClpP. We also characterized the solution phase behavior of small-molecule-activated ClpP using complementary structural techniques including methyl-TROSY NMR and small angle X-ray scattering (SAXS). Based on these data and extensive analysis of ClpP structures, we propose a consensus mechanism for ClpP activation by small molecules, wherein the reorganization of the electrostatic bonding networks at the axial entrance pores is necessary and sufficient for activation. Further activation is achieved by reduction of the structural flexibility of the N-terminal axial loops and of the ClpP cylinder in general.

## Results

**Binding mode of ACP1 to EcClpP.** To elucidate the structural basis for ClpP activation by ACP1[28], structures of NmClpP and EcClpP (Fig. 1a) in complex with ACP1 analogs were sought. While a structure of NmClpP with bound ACP1 was not achieved despite repeated attempts, a structure of the EcClpP+ACP1-06 complex was determined to 1.9 Å resolution (Fig. 1b, c, Table 1), containing a tetradecamer in the asymmetric unit (Chains A-N). Electron density for N-terminal residues that form the axial loops of EcClpP is unclear in all but one subunit (Chain B). In previously published structures, these axial loops are highly flexible and are usually disordered in the crystal in the absence of activators or through preclusion by crystal packing[23]. Unambiguous electron density was observed for a single ACP1-06 molecule in a pocket formed by subunits D and E, showing two different configurations of the compound (Fig. 2a, b). Both configurations were modeled in the electron density, in which the first configuration positions the trifluoromethylpyridine portion of the compound within a hydrophobic pocket (down configuration), while the second positions it out of the hydrophobic pocket exposed to solvent (up configuration) (Fig. 2a). The remaining 13 hydrophobic pockets show ambiguous electron density for ACP1-06. We modeled and refined all 14 ACP1-06 molecules in both up and down configurations at occupancy of 0.5, resulting in improved electron density for each ligand.

The binding modes of ACP1-06 (Fig. 2b–d) approximate those of ADEPs observed in crystal structures of different bacterial ClpPs (Fig. 2c–f)[6,15,18,19,21,40]. Note that compounds synthesized by our group are numbered as ACP1-YY and ADEP-YY, while compounds from studies by other groups are named as in the respective published papers. All protein contacts with ACP1-06 are nonpolar in nature except for two notable electrostatic bonds that occur between (1) the phenolic hydroxyl side chain of Y76 and the amide oxygen of ACP1-06, and (2) the guanidinyl side chain of R206 (the penultimate Arg in EcClpP) and a sulfonyl oxygen of ACP1-06 (Fig. 2b). The latter two interactions are present in both configurations of ACP1-06. In addition, R206 (Chain E) is stabilized by ionic bonding with E65 of an adjacent subunit (Chain D) (Fig. 3a, b).

In the down configuration, the trifluoromethylpyridine moiety of ACP1-06 occupies a hydrophobic cavity formed by L62, T93, and F96 (Chain D), and Y74, Y76, I104, L203, and L128 (Chain E) (Figs. 2b and 3b). This is the same cavity occupied by the ring of the exocyclic Phe residue of the different ADEP analogs cocrystallized with ClpP, such as in our NmClpP+ADEP-04[19] (Fig. 2e, f) or the EcClpP+ADEP1 structures[40] (Fig. 2c, d). By contrast, in the up configuration, the trifluoromethylpyridine moiety is rotated around the C–S bond and is solvent exposed while making van der Waals contacts with Y74, I104, F126, and L203 (Chain E) (Figs. 2b and 3b). The *gem*-dimethyl moiety stacks against the flat ring of Y74 (Chain E), while the extended aliphatic chain terminating with an *ortho*-chloro-substituted phenyl ring, sits in a nonpolar groove between two helices from adjacent subunits (Fig. 3b). This binding cleft is formed by L62 (Chain D), F63 (Chain D), A66 (Chain D), L37 (Chain E), V42 (Chain E), and the nonpolar portion of E40 (Chain E) (Figs. 2b and 3b).

In both ACP1-06- and ADEP1-bound structures of EcClpP, the intrasubunit salt bridge between R36 and E40 is found, where the nearby aliphatic tail of ADEP1 or the chloro-substituted phenyl ring of ACP1-06 forms a hydrophobic interaction with the side chain of E40 (Fig. 3b, c). The two activators are further stabilized in the hydrophobic site by two distinct hydrogen bond patterns. While ACP1-06 is secured through a solvent-exposed ionic interaction between its sulfonyl group and the C-terminal R206 residue (Fig. 3b), ADEP1 is held in place by two hydrogen

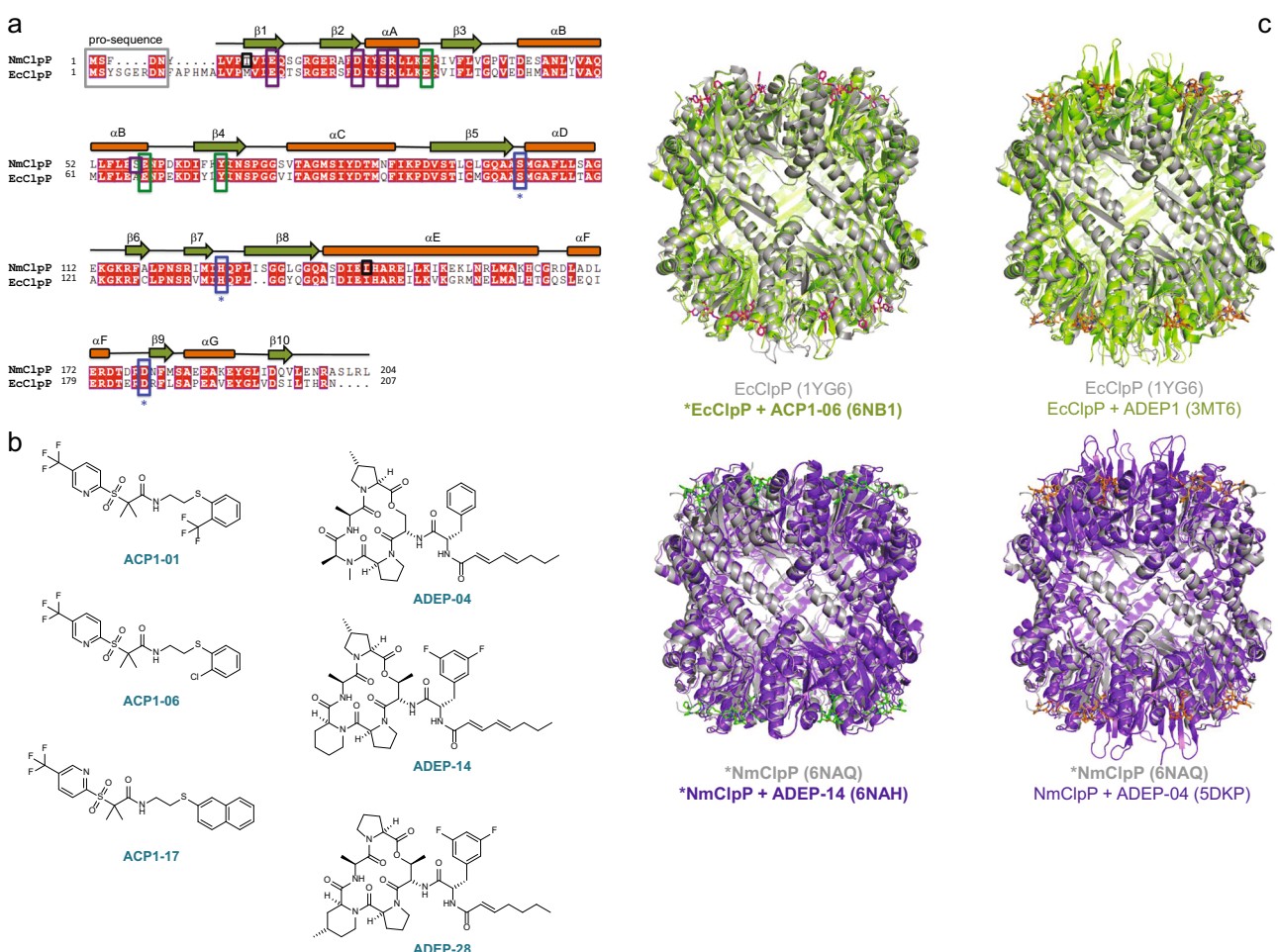

**Fig. 1** Sequence and structure of ClpP from *N. meningitidis* and *E. coli*. **a** Sequence alignment of ClpP from *N. meningitidis* (NmClpP) and *E. coli* (EcClpP). The pro-sequence is in the gray rectangle. Residues of the Ser-His-Asp catalytic triad are in the blue rectangles and marked with asterisks. Residues mutated in NmClpP (E31, E58) and EcClpP (E40, E67, Y76) as described in Fig. 4a, b are highlighted in green rectangles. Residues that participate in electrostatic interactions near the axial pore are enclosed in purple rectangles (E13, D23, S26, R27, and S57 in NmClpP). Residue S57 in NmClpP corresponds to A66 in EcClpP. Residues T10 and I144 of NmClpP mutated in spin-labeling NMR studies are enclosed in black squares. Secondary structures of NmClpP are shown on top of the sequence. **b** Chemical structures of the different compounds used in this work. **c** Crystal structures of ClpP showing global conformational changes upon ACP1 or ADEP binding. The structures determined in this study (indicated by asterisk and with labels bolded) of EcClpP with ACP1-06 (6NB1), apo-NmClpP (6NAQ), and NmClpP with ADEP-14 (6NAH)) are shown together with the structures of apo-EcClpP (1YG6[10]), EcClpP with ADEP1 (3MT6[40]), and NmClpP with ADEP-04 (5DKP; also solved by our group[19]) for comparison. Proteins are in cartoon representation, while ACP1 and ADEP compounds are in stick models. Apo structures of EcClpP and NmClpP are colored gray, while activator-bound structures are colored green (EcClpP) and purple (NmClpP+ADEP-14/ADEP-04). Axial loop ordering is observed in ADEP1- or ADEP-04-bound structures of EcClpP and NmClpP, respectively, while this is not observed in the ADEP-14 bound structure of NmClpP due to crystal packing (Supplementary Fig. 1d)

bonds with the hydroxyl group of Y76 secluded within the hydrophobic site (Fig. 3c). A solvent-mediated hydrogen bond between E65 and the carbonyl group of the N-acyl Phe side chain of ADEP1 further strengthens its interaction with EcClpP (Fig. 3c). This extra hydrogen bond, as well as the larger size of the cyclic depsipeptide ring, which has more surface area with which to form van der Waals interactions compared to the smaller trifluoromethylpyridine group of ACP1, may account for the generally tighter binding observed for ADEPs than ACP1s[28].

In the EcClpP+ACP1-06 structure, we found an unexplained electron density in all 14 subunits extending from the nucleophilic residue S111 of the catalytic triad, suggesting a covalent modification (Supplementary Fig. 1a, b). The density extends at approximately right angles in opposite directions away from the S111 side chains. Despite extensive efforts, it could not convincingly be fitted with peptides, acylketone products or various known serine protease inhibitor molecules.

**Binding mode of ADEP to NmClpP**. NmClpP has a shorter pro-sequence at the N-terminus, while having four extra residues at the C-terminus compared to EcClpP (Fig. 1a). Although NmClpP was expressed as a full-length protein carrying an N-terminal His$_6$-tag, we repeatedly observed lack of binding to the Ni-nitrilotriacetic acid resin. N-terminal sequencing of the purified protein revealed that the mature NmClpP starts at residue Y6 (Fig. 1a), indicating autoproteolysis to release its N-terminal pro-sequence ([1]MSFDN[5]).

Previously, we had determined the structure of NmClpP with ADEP-04[19]. In this study, we determined the structure of apo-NmClpP to 2.0 Å and NmClpP with bound ADEP-14 to 2.7 Å (Fig. 1b, c, Supplementary Fig. 1c, Table 1). The structure of apo-NmClpP contains a tetradecamer in the asymmetric unit and shows no clear density for any of the 14 N-terminal axial loops. Weak electron density is observed for the loops formed by residues G133-G137 in strand β8 of the handle region (Fig. 1a).

**Table 1 Data collection and refinement statistics**

|  | EcClpP+ACP1-06 | Apo NmClpP | NmClpP+ADEP-14 | NmClpP E58A | NmClpP E31A+E58A |
|---|---|---|---|---|---|
| *Data collection* |  |  |  |  |  |
| Space group | C 2 | P 2₁ | P 2₁ | P 2₁ | P 1 |
| Cell dimensions |  |  |  |  |  |
| *a, b, c* (Å) | 191.0, 101.1 155.2 | 98.6, 128.0,120.2 | 117.2, 196.0, 97.4 | 98.1, 119.4, 127.6 | 98.7,119.8, 127.9 |
| *α, β, γ* (°) | 90.0, 98.3, 90.0 | 90.0, 90.2, 90.0 | 90.0, 97.4. 90.0 | 90.00, 89.98, 90.00 | 90.02, 89.99, 90.02 |
| Resolution (Å) | 48.0-1.9 (2.00-1.90) | 51.4-2.0 (2.15-2.02) | 50.0-2.7 (2.75-2.70) | 50.0-2.4 (2.44-2.40) | 50.0-2.2 (2.24-2.20) |
| $CC_{1/2}$ (%) | 99.9 (70.2) | 99.6 (94.9) | 99.8 (55.9) | 99.9 (66.0) | 99.7 (76.9) |
| Completeness (%) | 98.3 (94.3) | 98.2 (96.8) | 98.7 (97.6) | 100.0 (99.7) | 95.4 (92.7) |
| Redundancy | 3.9 (3.4) | 3.9 (3.8) | 3.8 (3.8) | 3.7 (3.5) | 1.9 (1.8) |
| *Refinement* |  |  |  |  |  |
| Resolution (Å) | 48.0-1.9 | 51.4-2.0 | 49.6-2.7 | 42.0-2.4 | 46.1-2.2 |
| No. reflections | 223700 | 189962 | 168438 | 114987 | 283402 |
| $R_{work}/R_{free}$ | 20.9/24.4 | 20.7/24.6 | 19.6/25.1 | 19.3/25.2 | 21.5/24.3 |
| No. atoms |  |  |  |  |  |
| Protein | 20133 | 20048 | 37399 | 19461 | 38118 |
| Ligand | 812 | 0 | 1973 | 0 | 0 |
| Water | 923 | 633 | 691 | 897 | 1862 |
| B-factors (Å)² |  |  |  |  |  |
| Protein | 30.0 | 25.0 | 42.0 | 36.0 | 39.0 |
| Ligand | 61.2 | N/A | 44.8 | N/A | N/A |
| Water | 30.1 | 23.7 | 42.5 | 36.6 | 31.5 |
| r.m.s. deviations |  |  |  |  |  |
| Bond lengths (Å) | 0.010 | 0.008 | 0.009 | 0.009 | 0.010 |
| Bond angles (°) | 1.24 | 1.16 | 1.37 | 1.16 | 1.10 |

Values in parentheses are for highest-resolution shell

The asymmetric unit for the NmClpP+ADEP-14 crystal contains two tetradecamers (Supplementary Fig. 1d). No electron density was observed for residues 1–22 of all 28 subunits due to crystal packing (Fig. 1c, Supplementary Fig. 1d). In addition, the β8-strand (residues 130–137; Fig. 1a) is only partially visible in all subunits. ADEP-14 is bound to NmClpP in a similar configuration as ADEP-04 (Figs. 2e, f and 3e, f). Briefly, the difluorophenyl moiety of ADEP-14 occupies a hydrophobic pocket formed by Y67, L95, L97, and L119 of one subunit, and V49, L53, T84, and F87 of an adjacent subunit (Fig. 3e). The six-membered ring of the pipecolic acid moiety is solvent exposed and is stabilized by the phenyl ring of F117 and the hydrophobic side chains of L97, L119, and L196 (Fig. 3e). The additional methyl substitution on the *allo*-threonine residue of the depsipeptide ring (Fig. 1b) is solvent exposed. Like ADEP-04, ADEP-14 is secured in the highly complementary hydrophobic pocket by two hydrogen bonding interactions between the phenolic hydroxyl group of Y67, the amino group of the difluorophenylalanine residue, and the alanine carbonyl group in the depsipeptide ring, and a solvent-mediated hydrogen bond with E56 (Fig. 3e, f). The octadienoic acid side chain of ADEP-14 is located in the narrow hydrophobic channel formed by L53, F54, and S57 of one subunit, and R27, L28, E31, I33, F35, and Y67 of the neighboring subunit (Fig. 3e).

**ACP1- and ADEP-binding result in distinct allosteric effects on the ClpP barrel**. Using structures of apo- and compound-bound forms of EcClpP and NmClpP (Fig. 1c), we probed the allosteric effects that occur upon activator binding. As shown in Supplementary Fig. 2, activator binding acts as a wedge causing the lateral displacement of two adjacent subunits. The extent of global conformational change caused by ACP1-06 binding (rmsd = 0.84 Å relative to apo-EcClpP) is similar to that for ADEP-04 (rmsd = 0.73 Å relative to apo-NmClpP), but less than that for ADEP-14 (rmsd = 2.47 Å relative to apo-NmClpP). Interestingly, the direction of displacement is different between the two classes of activators (Supplementary Fig. 2 and Supplementary Movies 1–4). Between the two ADEPs, ADEP-14 causes a greater overall

structural perturbation to the NmClpP cylinder (compare Supplementary Movies 3 vs. 4). ADEP-binding results in an expansion of the apical surface of NmClpP accompanied by constriction at the equatorial region. The pivot for this motion is in the handle region composed of helix αE and strand β8. This phenomenon is observed in all ADEP-bound ClpP structures known to date, with varying degrees of compaction of the ClpP cylinder[15,18,19,23,40]. By contrast, ACP1-06 binding to EcClpP causes an inward movement of all subunits resulting in tightening of the ClpP cylinder (Supplementary Movie 1). Thus, the structures show that ACP1s and ADEPs activate ClpP by inducing distinct allosteric effects.

In the activator-bound EcClpP and NmClpP structures, the ring–ring interface electrostatic bonds that stabilize the tetradecamer remain conserved despite the conformational changes (Supplementary Fig. 3). Moreover, notwithstanding the global structural change upon activator binding, the Ser-His-Asp catalytic triads of EcClpP and NmClpP maintain catalytically competent geometries as only minor Cα backbone shifts occur near the active site (Supplementary Fig. 1a–c). Analysis of all existing ACP1- and ADEP-bound structures of ClpP shows that compound binding also results in the contraction of the equatorial region (Supplementary Fig. 4).

We quantified the compaction of the ClpP cylinder upon ACP1 or ADEP binding by measuring the volumes of the respective catalytic chambers (Supplementary Fig. 4). ACP1-06 binding causes a ~5% decrease in the catalytic chamber volume relative to apo-EcClpP. ADEP1-binding to EcClpP results in a similar decrease in catalytic chamber volume. This is also observed for other activator-bound ClpP structures such as ADEP-14-bound NmClpP, ADEP-bound BsClpP, and the ADEP-bound MtClpP1P2 heterooligomeric complex (Supplementary Fig. 4).

**ClpP activation results in the reorganization of the electrostatic bonding network at the axial pores**. In the structure of apo-NmClpP, two electrostatic bonds near the axial pore stabilize the interface of any two adjacent subunits (Fig. 3d,

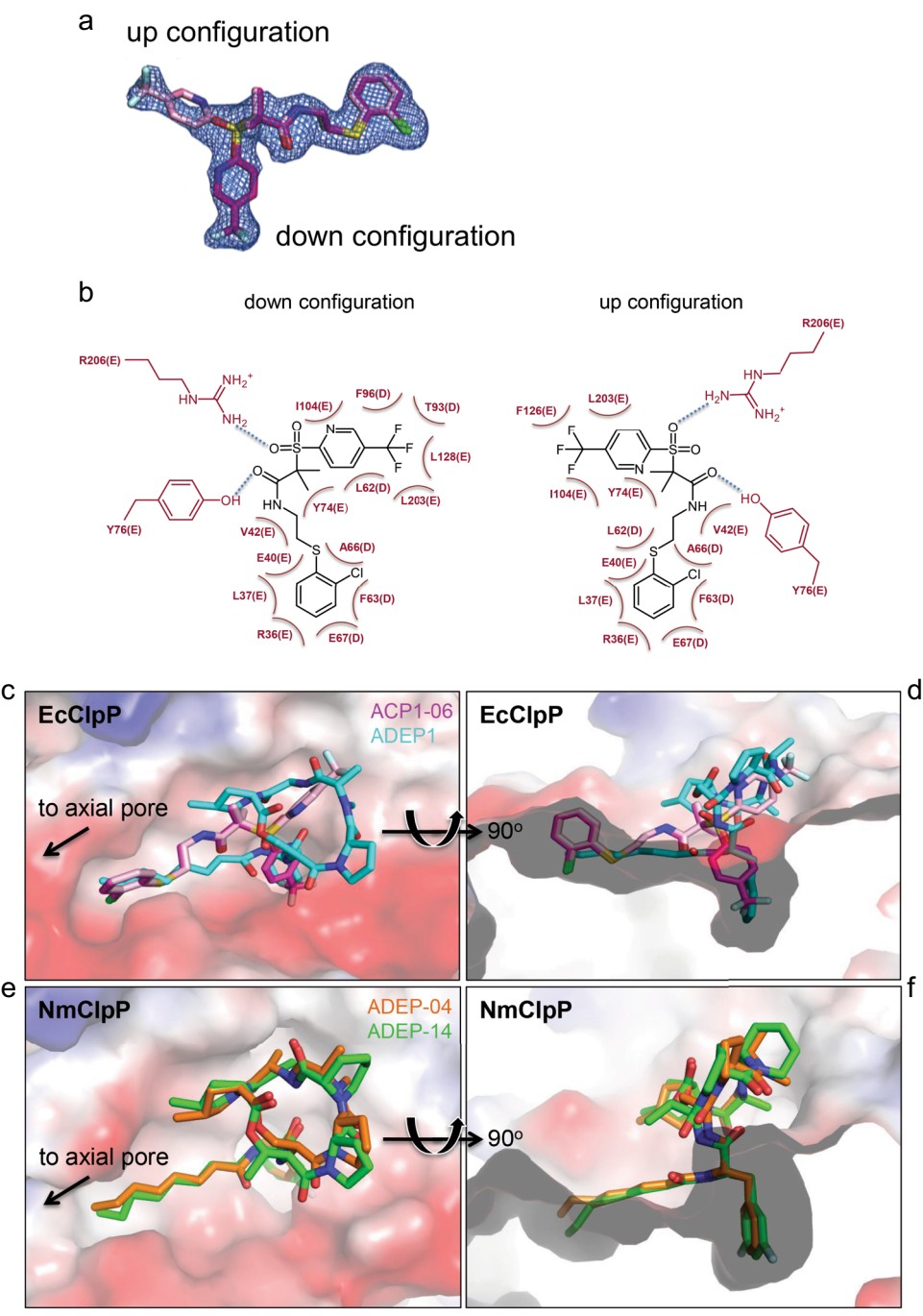

**Fig. 2** Binding modes of ACP1 and ADEP in the hydrophobic pocket of ClpP. **a** Omit map contoured at 1 σ showing the up and down configurations of ACP1-06. **b** Two-dimensional plots showing EcClpP residues that interact with ACP1-06 via hydrogen bonding and hydrophobic interactions. **c**, **d** Surface representations of the hydrophobic pocket of EcClpP showing the binding modes of ACP1-06 and ADEP1. The protein surface is colored according to its electrostatic potential with positive in blue and negative in red. ACP1-06 is shown in magenta and pink and ADEP1 in cyan. In (**d**), a cut view of the binding pocket is displayed highlighting the hydrophobic pocket, which accommodates the phenylalanine moiety of ADEP1 and the trifluoromethylpyridine group of ACP1-06 in the down configuration. **e**, **f** Surface representations of the NmClpP hydrophobic pocket with bound ADEP-04 and ADEP-14

Supplementary Fig. 5). First, the negatively charged E58 carboxylate group of one subunit forms an ion pair with the positively charged R27 guanidinium group of the adjacent subunit. Second, the S57 hydroxyl and E31 carboxylate groups of two adjacent subunits form a hydrogen bond. Upon ADEP binding, these two noncovalent bonds break, while the ionic bond between R27 and E31 of the same subunit is shortened, i.e. strengthened (Fig. 3e, f). In apo-EcClpP, only one ionic bond between E67 and R36 links

two adjacent subunits, since the equivalent residue to S57 of NmClpP is an alanine in EcClpP and is thus unable to form a hydrogen bond with E40 (Fig. 3a). As in NmClpP, binding of ACP1 or ADEP1 to EcClpP eliminates the intersubunit E67-R36 ionic bond and gives rise to a stronger intrasubunit ionic bond between R36 and E40 (Fig. 3b, c, Supplementary Fig. 5).

To further verify these observations, we designed NmClpP and EcClpP point mutants that lead to loss of the above intersubunit

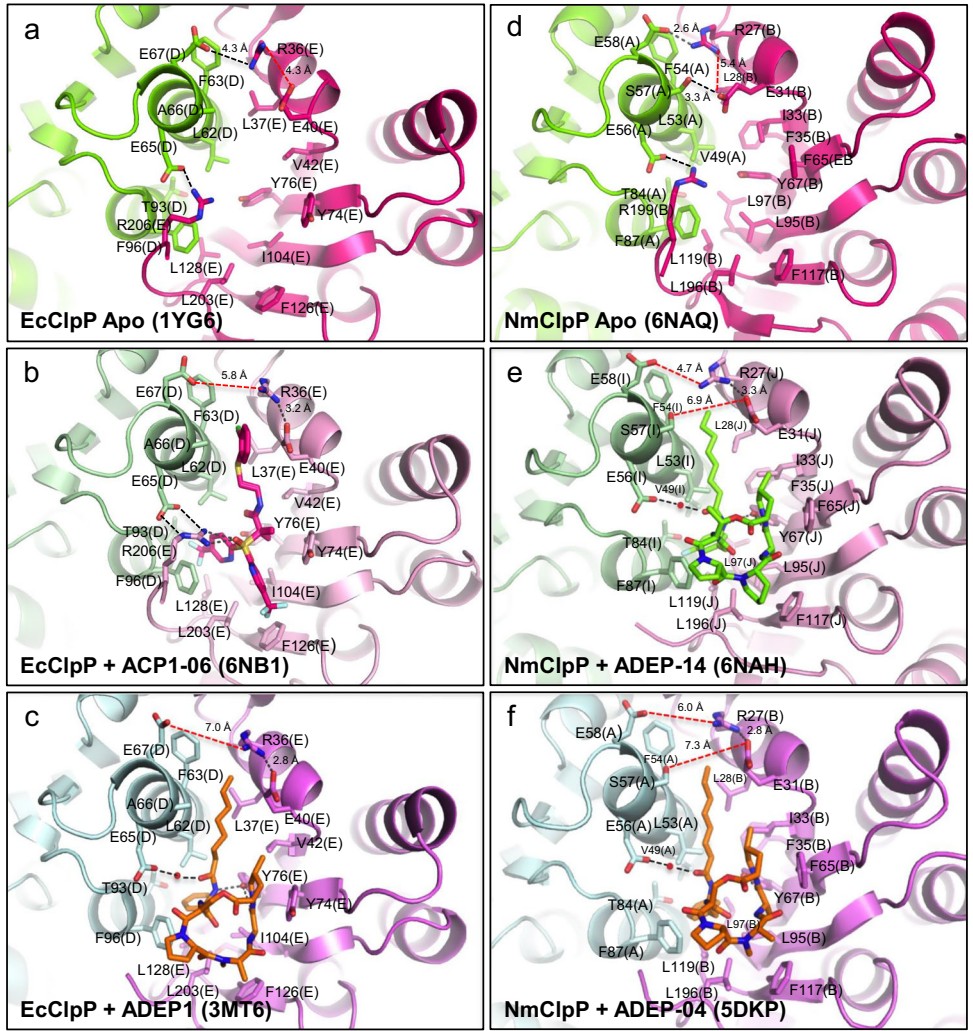

**Fig. 3** Close up view of the activator binding site in ClpP. **a–c** The interface between two subunits of apo-EcClpP, EcClpP+ACP1-06, and EcClpP+ADEP1. **d–f** The interface between two subunits of apo-NmClpP, NmClpP+ADEP-14, and NmClpP+ADEP-04. In all the panels, distances between residues discussed in the text are indicated by dashed lines. If the distance is ≤4.3 Å, then the dashed line is in black, otherwise the dashed line is in red. 4.3 Å is the distance observed in apo-EcClpP (1YG6) between the side chains of E67(D) and R36(E) (Fig. 3a)

electrostatic bonds. As shown in Fig. 4a, while WT NmClpP was unable to degrade the protein casein, the NmClpP E58A mutant was found to degrade casein at a higher rate than the E31A mutant. Importantly, the double mutant E31A + E58A had an even higher activity than the single mutants. The extent of activation of the double mutant NmClpP is similar to that observed in the presence of 1 μM ADEPs or 10 μM ACP1s (Fig. 4a). Similar behavior was observed for EcClpP with similar mutations (E40A and E67A; Fig. 4b). The respective EcClpP double mutant is not soluble and could not be tested.

Subsequently, we determined X-ray structures of NmClpP E58A and NmClpP E31A + E58A mutants (Table 1). The catalytic sites in both mutants are not perturbed (Supplementary Fig. 1e). In the NmClpP E58A single mutant, the mutation abolishes the intersubunit E58–R27 ionic bond and gives rise to a stronger intrasubunit R27–E31 interaction, while the hydrogen bond between S57 and E31 remains (Fig. 4c). A similar "wedge effect" is thus observed in the structure, as shown by the subtle global conformational change in the NmClpP mutant's Cα backbone (rmsd = 0.33 Å relative to WT NmClpP) (Supplementary Movie 5). Mutation of both E31 and E58 to alanine to generate the double mutant NmClpP E31A + E58A (rmsd = 0.29

Å relative to WT NmClpP) abolishes the two stabilizing intersubunit electrostatic interactions, as well as, the intrasubunit R27–E31 ionic bond present in the NmClpP E58A single mutant (Fig. 4d and Supplementary Movie 6) and in ADEP-bound NmClpP (Fig. 3e, f). Thus, the NmClpP E31A + E58A double mutant is unlike ADEP-bound NmClpP with the eliminated intrasubunit ionic bond but is nonetheless an activated protein with intrinsic proteolytic activity comparable to that of small-molecule-activated ClpP (Fig. 4a). Like ADEP-bound NmClpP, both NmClpP single and double mutants have reduced catalytic chamber volumes due to ClpP barrel compaction (Supplementary Fig. 4).

Such reorganization of the bonding network is observed for all activated ClpP structures available in the PDB, whether in the presence of small-molecule activators or due to specific mutations (Supplementary Fig. 6). For example, small-molecule activator binding to *B. subtilis* ClpP (BsClpP), *S. aureus* ClpP (SaClpP), *M. tuberculosis* ClpP (MtClpP), and *Homo sapiens* ClpP (HsClpP) abolishes one or two intersubunit electrostatic bonds near the axial pore, and gives rise to a stronger, intrasubunit ionic interaction (Supplementary Fig. 6a–e, g–l)[6,15,18,21,39]. In MtClpP2, the intersubunit ionic bond equivalent to the

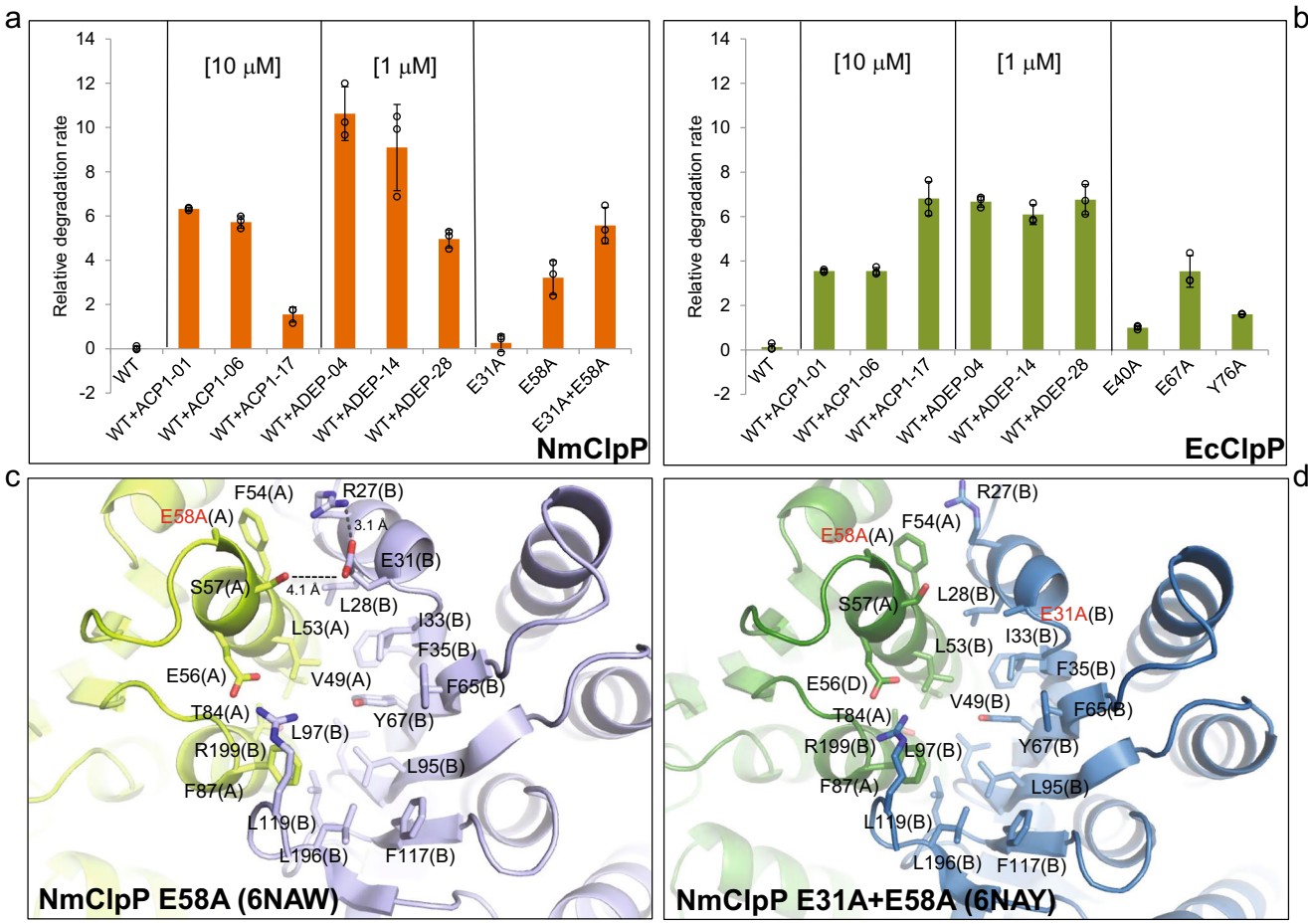

**Fig. 4 Activating mutations in NmClpP and EcClpP. a, b** Comparison of the proteolytic activities of mutant EcClpP and NmClpP with compound-activated ClpP using casein-FITC as model substrate. Compound structures are shown in Fig. 1b. All experiments were carried out 3 times. Source data are available in Supplementary Data 1. **c** The interface between two subunits in NmClpP E58A mutant. **d** The interface between two subunits in NmClpP E31A + E58A mutant

E58–R27 interaction in NmClpP does not exist[18]. The equivalent residue for E58 of NmClpP is L66 in MtClpP2 and is unable to participate in an ionic interaction with K35 of MtClpP2 (equivalent to R27 of NmClpP). Instead, an intersubunit hydrogen bond between S65 and E39 stabilizes the interface (Supplementary Fig. 6g, h). ADEP binding to MtClpP2 breaks the S65–E39 hydrogen bond and strengthens the intrasubunit ionic bond between K35 and E39[18].

Interestingly, even the activated SaClpP Y63A variant[41], with the activating mutation found in the center of the hydrophobic site, and, therefore, not equivalent to the activating mutations of NmClpP presented here, supports our proposed model of the reorganization of the hydrogen bonding network around the axial pore upon ClpP activation (Supplementary Fig. 6f). In the structure of the SaClpP Y63A mutant, the distance between the intersubunit ion pair (Q54–R23) is increased, while the intrasubunit salt bridge (R23–D27) is strengthened (Supplementary Fig. 6d, f). We also generated the equivalent Y63A mutation in both EcClpP and NmClpP. The NmClpP Y67A mutant was insoluble, while the EcClpP Y76A had an activity slightly higher than the E40A mutant, but lower than that of the E67A mutant (Fig. 4b).

**ClpP activation results in reduction in structural heterogeneity of the N-terminal axial loops.** As discussed above, in the ordered axial loop of the NmClpP+ADEP-04 complex forming the β1–β2

hairpin turn, ADEP binding releases R27 from an intersubunit ionic bond with E58 and strengthens the intrasubunit ionic bond with E31 of helix αA (Fig. 5a). Consequently, R27 forms an ionic bond with D23 of strand β1 of the axial loop. This stabilizing interaction is observed in all activator-bound structures of ClpP where the axial loops are ordered (Fig. 5a–e).

For the EcClpP+ACP1-06 structure, the axial loops (residues 14-31) are partially ordered in the crystal with only 1 out of 14 axial loops forming the β1–β2 hairpin turn (Fig. 1c—complete ordering seems to be partly prevented by crystal packing effects). In the ordered axial loop of subunit A, release of R36 from the intersubunit ionic bond with E67 allows it to form a stronger intrasubunit ionic bond with E40 of helix αA (Fig. 5f). However, there is no stabilizing ionic interaction between E22 of strand β1 and R36 of helix αA, likely due to partial ordering (Fig. 5f). An additional hydrogen bond between D32 of strand β2 and S21 of helix αA anchors the axial loop to the EcClpP core domain (Fig. 5f). Similarly, the axial loops in the *Enterococcus faecium* ClpP (EfClpP)-ADEP4 structure are only partially ordered[21]. The conserved hydrogen bond between the Arg residue of helix αA and a negatively charged residue of strand β1 is not seen in the partially ordered EfClpP axial loops, although EfClpP has potential hydrogen bond forming residues on strand β1 (T6), and on the loop connecting strands β1 and β2 (E9, Q10, S11, S12, E15) (Fig. 5g).

In addition to conserved electrostatic interactions, extensive hydrophobic contacts with the ClpP head domain stabilize the

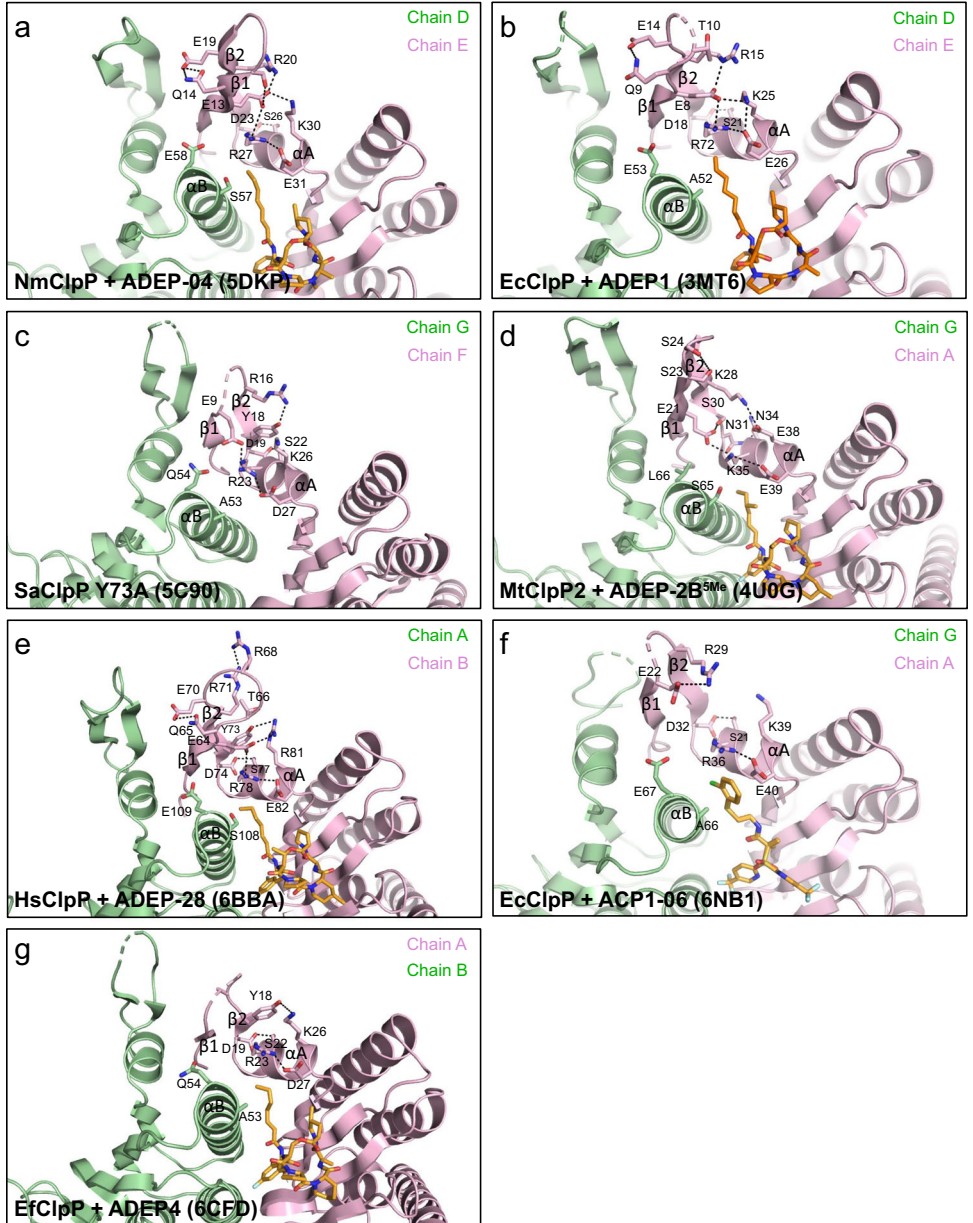

**Fig. 5** Ordering of the N-terminal axial loops in activated ClpP. **a–g** The relevant interactions promoting axial loop ordering in ClpP-activating compound complexes are highlighted

axial loops. In EcClpP+ACP1-06, nonpolar N-terminal residues of strand β1 and the preceding structured coil participate in hydrophobic interactions with nonpolar residues of helix αA of the same subunit, and on the hydrophobic faces of helices αA′ and αB′ of a neighboring subunit (Supplementary Fig. 7a). The nonpolar residues on helices αA, αB′, and β3′ constitute a continuous hydrophobic patch on the circumference of the axial pore (Supplementary Fig. 7b)[15].

To get a clearer picture of the conformational heterogeneity of the ClpP axial loops, methyl-TROSY NMR experiments were then carried out. Initially, a single cysteine mutation was introduced in the axial loops of NmClpP(T10C). Uniformly deuterated protein was produced and subsequently reacted with $^{13}$C-methyl-methanethiosulfonate (MMTS). This results in the attachment of a single NMR visible $^{13}CH_3$–S group to the cysteine side chain, leading to formation of an S-methylthio-cysteine (MTC) residue[42]. This method provides a facile way of monitoring the structure and dynamics of large complexes in

solution. Monitoring the NMR correlations of the attached spin probe in free, activator-bound or mutated forms of the enzyme provides a readout of the solution conformation of the axial pores.

Initially, control experiments were performed to ensure that the introduction of the T10MTC moiety does not perturb the structure of NmClpP. Briefly, $^1$H–$^{13}$C heteronuclear multiple quantum coherence (HMQC) correlations of WT and T10MTC NmClpP labeled $^{13}CH_3$ at the side chain of ILVM residues in an otherwise fully deuterated background were compared and found to be virtually identical. Subsequently, $^1$H–$^{13}$C HMQC correlations of NmClpP T10MTC were obtained in different states (Fig. 6a). The apo-form (blue contours) has a large number of correlations, indicating structurally heterogeneous axial loops. Addition of two-fold molar excess of ADEP-28 (activity given in Fig. 4a) over monomeric ClpP significantly reduced the number of correlations (red contours), indicative of rigidification or structural ordering. By contrast, ACP1-17 binding (two-fold

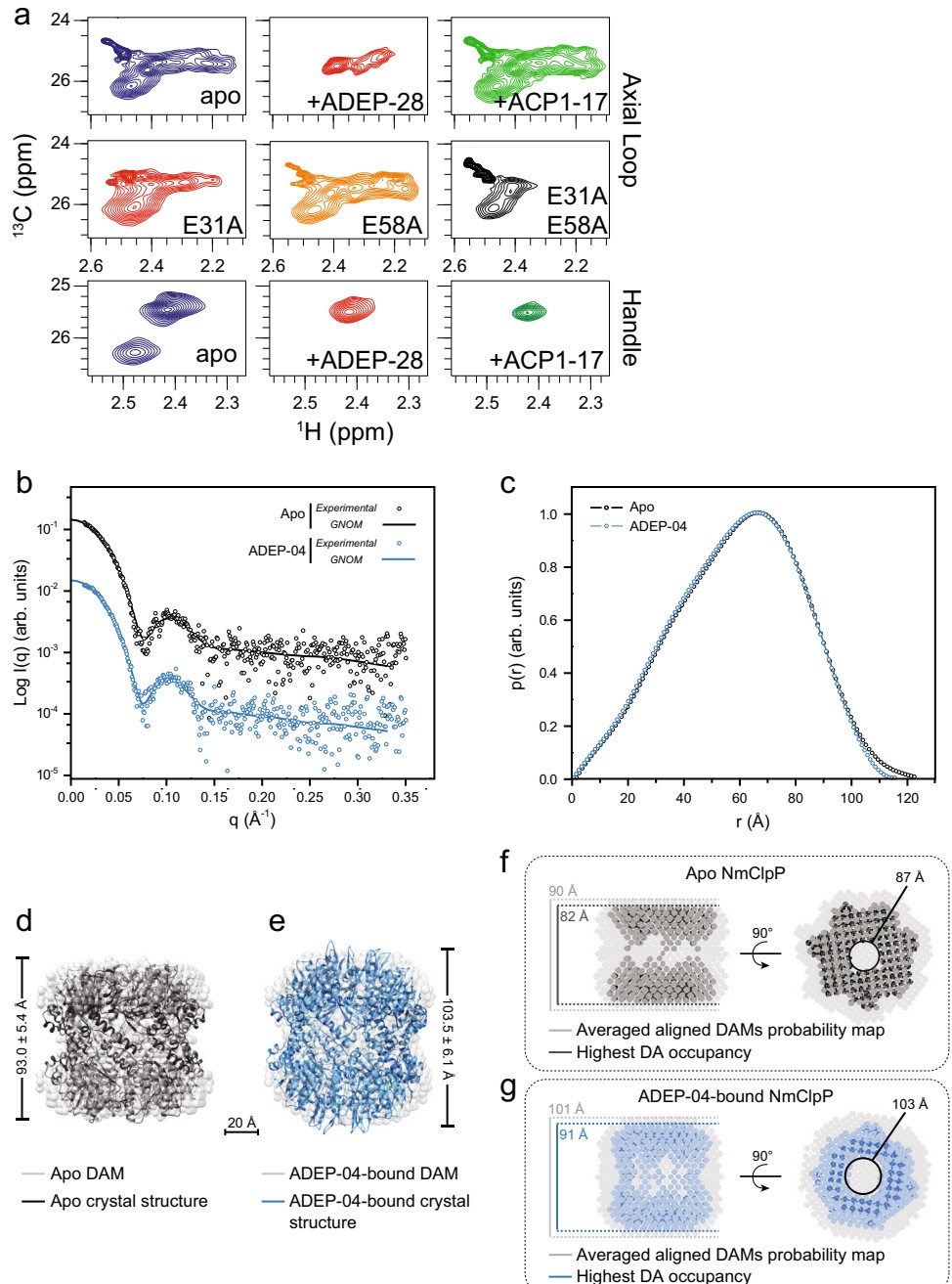

**Fig. 6** Analysis of NmClpP dynamics by NMR and of the protease solution structure by SAXS. **a** $^1$H–$^{13}$C HMQC spectra of NmClpP labeled with MMTS to produce single $^{13}$CH$_3$ probes on axial loop (positions 10) and handle helix (position 144). Spectra were recorded in apo-, ADEP-28- and ACP1-17-bound forms, as well as, for constructs bearing activating mutations. NmClpP protomer concentration ranged between 200–250 μM. Compounds were at two-fold molar excess over protomer concentration. **b** Scattering curves of NmClpP in the absence (black) and presence (blue) of ADEP-04. Symbols represent experimental data; solid lines represent the fitting of GNOM curves. The values for the NmClpP+ADEP-04 curve were divided by 10 for comparison purposes. **c** Pair distance distribution functions, p(r), of NmClpP determined by the GNOM software. Apo NmClpP is displayed in black, whereas NmClpP-ADEP-04 is shown in blue. **d**, **e** Most probable dummy atom models (DAMs) for apo-NmClpP and NmClpP+ADEP-04 are shown as gray dots. The fitting of DAM scattering profiles to experimental data are shown in Supplementary Fig. 9b. Apo-NmClpP (black) and NmClpP+ADEP-04 (blue) crystal structures were superimposed onto the DAMs using the SUPCOMB program[65]. DAMs averaged axial height based on ten independent DAMMIN runs are displayed next to each model. The scale bar is shown at the bottom. **f**, **g** DAMs probability maps (gray dots) derived from the average of all models generated by the DAMMIN program[62] (mean normalized spatial discrepancy, NSD, of 0.745 ± 0.033 for apo-NmClpP and 0.708 ± 0.027 for ADEP-04-bound NmClpP; values close to 0 refer to ideally superimposed structures and values higher than 1 to significantly different ones) and regions of the highest DAs occupancy are shown for apo-NmClpP (black) and ADEP-04-bound NmClpP (blue) in two different views. The heights for both averaged probability maps and highest DA occupancy maps are given. The circumferences of the axial pores for the highest DA occupancy maps are also given. 3D structures and DAMs were rendered using the UCSF Chimera software (https://www.cgl.ucsf.edu/chimera/)

molar excess over monomeric ClpP; activity given in Fig. 4a) results in undetectable changes in the observed correlations (green contours), implying that the axial loops are unaffected.

This approach was also exploited to monitor the effect of activating mutations (Fig. 4a, b) on the axial loops. In these experiments, the activating mutations were introduced in the background of the T10C (labeling) mutation. As shown in Fig. 6a, correlations of the NmClpP T10MTC/E31A mutant (red contours) is slightly less heterogeneous than the pseudo WT form, while those of the NmClpP T10MTC/E58A mutant (orange contours) are virtually unchanged. The simultaneous presence of both activating mutations (NmClpP T10MTC/E31A + E58A) has a much more dramatic effect than single mutations and removes a large number of correlations corresponding to the axial loops (black contours). This is consistent with the observation that the double mutant is more active than the single mutants (Fig. 4a).

**ClpP activation results in reduction in conformational heterogeneity of the handle region.** The same NMR-based approach was used to probe the effect of activator binding and mutations on the handle region. An I144MTC (helix αE) mutant of NmClpP was prepared and studied by NMR in apo-, ADEP-28-, and ACP1-17-bound forms. The apo-form (blue contours, Fig. 6a) exhibits a pair of peaks, indicating that the handle region is associated with a pair of coexisting conformations, as seen in our previous work on EcClpP[4]. Interestingly, addition of ACP1 and ADEP leads to the disappearance of one of the peaks. Given that the activator binding site is distal to the NMR spin probe, it appears that ACP1 or ADEP binding allosterically selects for one of the conformations in the handle region. Alternatively, activators might be able to bind both forms but induce a change to a single state.

Magnetization exchange experiments with mixing delay periods of 100, 200, 300, 400, 500, and 600 ms at 40 °C were unable to detect any interconversion between the conformers observed for the handle region for WT NmClpP. This indicates that the exchange process is too slow for characterization by NMR.

**SAXS demonstrates activator-induced ClpP conformational changes in solution.** To further probe the oligomeric state and structural changes upon compound binding, apo-NmClpP and NmClpP+ADEP-04 samples were characterized by SAXS (Fig. 6b–g and Supplementary Fig. 8). Final merged curves are shown in Fig. 6b, and the SAXS profiles obtained resembled those of hollow structures[43]. No significant changes were found in terms of overall folding (Supplementary Fig. 8), radius of gyration ($R_g$) and oligomeric state when ADEP-04 was added to NmClpP (Supplementary Fig. 8c). In order to further analyze the NmClpP SAXS profiles and generate solution ab initio structures, the GNOM program was used to construct pair distance distribution functions, $p(r)$. Apo-NmClpP $p(r)$ revealed a subtle right-shift and larger maximum dimension ($D_{max}$) in comparison to ADEP-04-bound NmClpP (Fig. 6c and Supplementary Fig. 8c). Subsequently, dummy atom models (DAMs) were generated using SAXS data to visually analyze NmClpP solution structures (Fig. 6d–g). Ten models were generated for apo-NmClpP and ADEP-04-bound NmClpP, and the most probable ones were chosen. These models showed that, overall, the two NmClpP solution structures are similar (hollow cylinders). However, when superimposed with the corresponding crystal structures, differences that agree with the high-resolution structures were easily observed (Fig. 6d, e). A measure of the axial heights of all ten DAMs for apo- and ADEP-04-bound NmClpP resulted in values of 93.0 ± 5.4 Å for the apo-form and 103.5 ± 6.1 Å for the ADEP-04-bound form. To further corroborate this observation, averaged

DAMs probability maps and highest DAs occupancy maps (Fig. 6f, g) were analyzed. The axial heights displayed an increase of about 10 Å for ADEP-04-bound NmClpP in comparison with apo-NmClpP. Moreover, ADEP-04-bound NmClpP displayed a larger axial pore circumference than apo-NmClpP (Fig. 6f, g). Thus, despite the low resolution of the SAXS technique, the results suggest that ADEP-04 binding to NmClpP does not affect the oligomeric state of NmClpP but causes an expansion of the axial pores and increased occupancy at the top and bottom parts of the NmClpP+ADEP-04 DAM (Fig. 6e, g) compared to apo-NmClpP. This likely reflects the conformational changes leading to the decreased heterogeneity of the structure of the N-terminal axial loops of NmClpP observed by X-ray and NMR.

## Discussion

The high-resolution crystal structures of ClpP in complex with ACP1 and ADEP activators presented here show that, while the two classes of compounds share a common binding site, their mechanisms of activation are dissimilar in both magnitude and detail (Supplementary Fig. 2 and Supplementary Movies 1–4). ADEP binding induces a more pronounced conformational change in ClpP, with the magnitude of axial pore expansion and equatorial constriction being greater than those caused by ACP1 binding. In ADEP-bound complexes of NmClpP, the head domain moves away from the central 7-fold axis, while the handle region moves toward the axis as the barrel constricts at the equator. By contrast, ACP1 binding to EcClpP tightens the protease barrel by drawing all 14 subunits inwards towards the long axis. These conformational changes in ACP1- and ADEP-bound structures result in reduced catalytic chamber volumes (Supplementary Fig. 4) and wider axial pores that can facilitate substrate entry and increase proteolytic activity. Importantly, we demonstrated that, specific mutations which abolish electrostatic interactions lining the axial pore (Fig. 4a, b), mimic the wedge-like action of these small-molecule activators.

Our analysis also revealed the molecular details that link axial loop ordering to catalytic activation (Fig. 7a). In activated ClpP, elimination of an intersubunit ion pair involving a conserved Arg/Lys residue on helix αA and Glu/Gln residue on helix αB′ frees the Arg/Lys residue to form an intrasubunit salt bridge with a conserved Glu/Asp residue on strand β1 of the axial loop, while the interaction of Arg/Lys with a conserved Glu/Asp residue on the same helix αA is strengthened (Fig. 7a). In addition, a conserved hydrogen bond between an Asp/Asn residue on strand β2 and a Ser/Asn residue on helix αA further fastens the N-terminal β-hairpin structure to the ClpP core (Fig. 7a). In some ClpPs, a second intersubunit interaction between a Ser on αB′ and the Glu/Asp on αA is present and is broken or weakened upon activator binding (Fig. 7a). Conserved extensive sets of hydrophobic interactions between nonpolar residues of the ordered N-terminus preceding strand β1, and those of helices αA, αB′, and β3′ complete the stabilizing contacts of the ordered axial loops with the ClpP cylinder (Supplementary Fig. 7a). The net effect of axial loop ordering is the retraction of disordered N-terminal sequences that plug the axial pore, forming a collar that delimits the axial openings of ClpP. Many of the residues involved in these interactions are conserved in ClpP from different species (Supplementary Fig. 9). Further activation can then be achieved by reducing structural heterogeneity around the axial pores and the equatorial region, leading to a fully active conformer of ClpP (Fig. 7b). Thus, the unified mechanism we propose (Fig. 7b) articulates both the minimum requirement for activation (electrostatic network reorganization around the axial pores) and the existence of an activation gradient for ClpP that encompasses different conformational states.

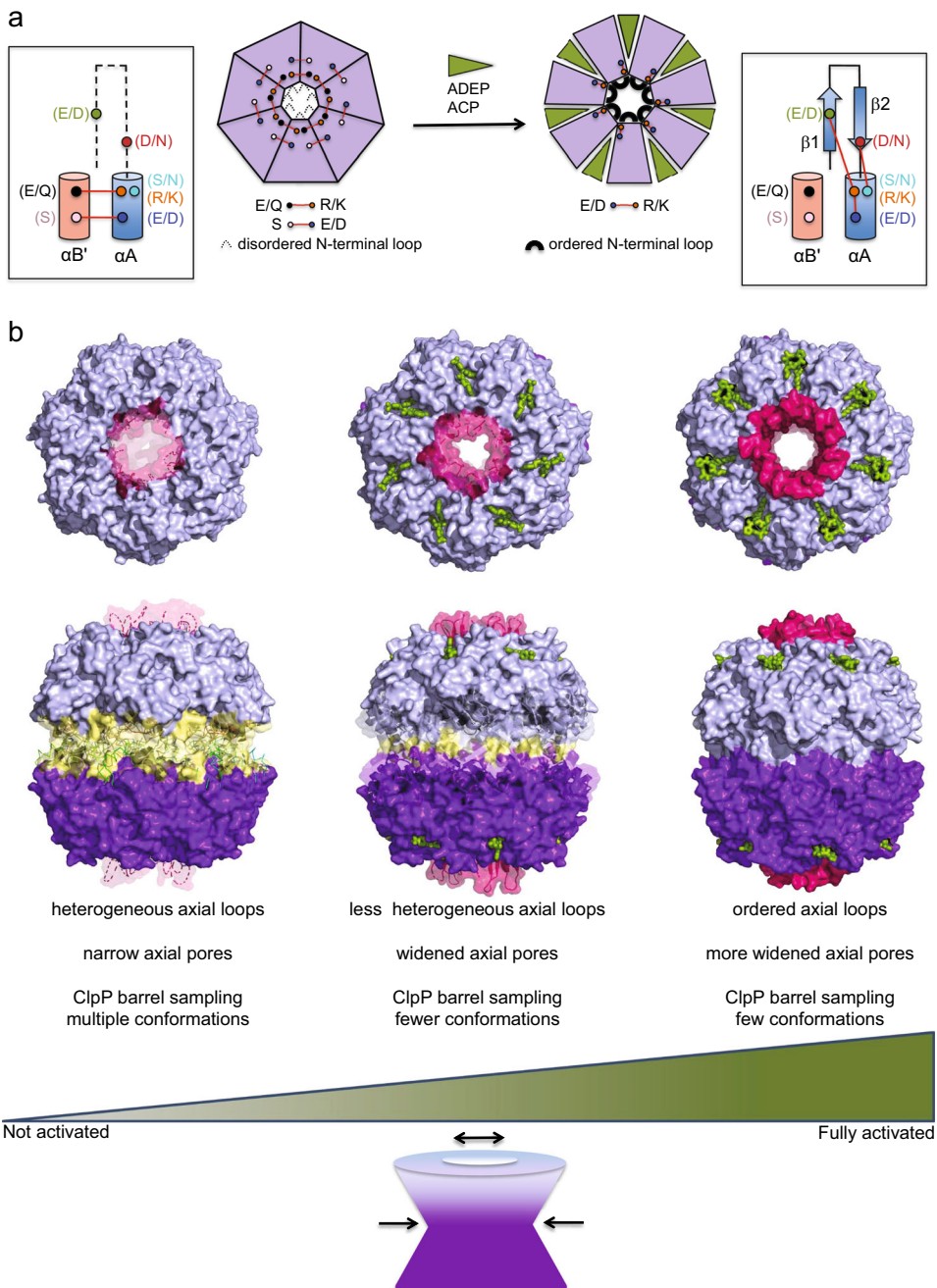

**Fig. 7** Mechanism of ClpP activation by small molecules. **a** Schematic diagram showing mechanism of axial loop ordering. Small-molecule activator binding to hydrophobic binding pockets acts as a wedge between adjacent subunits, abolishing one or two intersubunit electrostatic bonds near the axial pore typically formed between two neighboring subunits (Glu/Gln in αB' and Arg/Lys in αA; Ser in αB' and Glu/Asp in αA). This gives rise to a stronger intrasubunit ionic bond between the Arg/Lys and a Glu/Asp in αA, as well as, to interactions between the Arg/Lys in αA and Glu/Asp in β1 and Ser/Asn in αA and Asp/Asn in β2. The latter interactions promote axial loop ordering. **b** Proposed activation spectrum for ClpP, beginning in the apo state. In this state (left structures), axial loops are disordered (broken pink loops), and there is significant heterogeneity in the equatorial region (various conformations of secondary structures with yellow coloring indicates structural heterogeneity). Axial loop disorder plugs the axial pores. Weaker activators such as some ACP1s (green sticks in middle structures) decrease axial loop structural heterogeneity, widen the axial pores, and rigidify the ClpP structure to a significant extent (decreased yellow coloring at the equatorial region), but to a lesser extent than that by stronger activators such as ADEPs (green sticks in rightmost structures). In the fully activated state, axial loops are ordered (solid pink surface), and ClpP structural heterogeneity is further reduced (solid surface for the ClpP barrel). Mutations that disrupt bonding networks near the axial pores result in ClpP species whose activities lie somewhere in the activation spectrum and possess structural features in common with small-molecule-activated ClpP, including less conformational heterogeneity in axial loops and handle regions relative to the apo state. Structures used to depict states in the activation spectrum have the following PDB accession codes: 1TYF (EcClpP) for non-activated apo state, 6NB1 for intermediate state (EcClpP + ACP1-06), and 3MT6 (EcClpP + ADEP1) for fully activated state. Ordered axial loops from 3MT6 were superimposed and then shown as broken lines with transparent surface to depict heterogeneity in the apo state, or as partially ordered structures with slightly darker surfaces in the intermediate state

The previously accepted mechanism for ClpP activation emphasizes three structural features induced by activator binding: (1) formation of ordered axial loops (β-hairpin turns) from disordered N-terminal loop sequences, whose first few residues are hydrophobic and plug the axial pore in the apo state (dissolution of hydrophobic plug)[31]; (2) anchoring of the β-hairpin turns via their hydrophobic N-termini to the hydrophobic patch lining the circumference of the axial pores with the alkyl/aryl groups of the activators serving as hydrophobic nucleation sites to trigger the retraction of the β-hairpin turns away from the center of barrel (hydrophobic pull)[31]; and (3) rigidification of the equatorial handle region, thereby constraining ClpP to its active form (conformational selection)[31,34].

Our observations on ACP1-ClpP complexes indicate that contrary to feature 1 of the current mechanism, complete dissolution of the hydrophobic plug by formation of all 14 axial loops is not required for activation. The structure of EcClpP-ACP1-06 complex (Fig. 5f) is evidence to this claim, in conjunction with the result of our NMR studies on NmClpP with ACP1-17. Rather, we find that mere reorganization of electrostatic interactions near the axial pore, is the critical event preceding hydrophobic plug dissolution (feature 1) and the hydrophobic pull (feature 2) and is necessary and sufficient for ClpP activation (Figs. 3 and 7). This new insight thus resolves the obfuscation of the true significance of axial loop ordering in the context of activation, imposed by invisible or only partially ordered axial loops often seen in structures of activated ClpP complexes[15,21,39,44].

## Methods

**Plasmids and subcloning.** The *N. meningitidis clpP* gene was PCR amplified from *N. meningitidis* MC58 genomic DNA with either Pfu (Fermentas) or Phusion (Thermo Fisher) DNA Polymerase. The PCR products were inserted into p11, a pET vector (Novagen) modified to add an N-terminal 6x-His tag followed by a tobacco etch virus (TEV) protease recognition sequence. The plasmid places the gene under the control of isopropyl β-D-1-thiogalactopyranoside (IPTG) inducible promoter. The construct was verified by Sanger sequencing. Mutants of NmClpP were constructed using designed primers and standard mutagenic PCR reactions. The plasmid pET9a EcClpP used for protein overexpression was a generous gift from Dr. John Flanagan (Pennsylvania State University College of Medicine, USA).

**Protein expression and purification for biochemical assays and X-ray structure determination.** ClpP constructs were expressed in the BL21(DE3)1146D strain, which lacks the gene for endogenous EcClpP. EcClpP and NmClpP were expressed and purified generally as previously described for EcClpP[45]. Briefly, an overnight culture of BL21(DE3)1146D cells transformed with the respective ClpP plasmid was inoculated at a 1:100 dilution into LB media supplemented with ampicillin. The culture was agitated for 3 h at 37 °C to an $OD_{600}$ of 0.5 to 0.7. IPTG was added to a final concentration of 1 mM, and the culture was incubated for another 4 h. Cells were harvested by centrifugation and resuspended in buffer A (50 mM TrisHCl, pH 7.5, 150 mM KCl, 10% glycerol, and 1 mM DTT), and lysed using a French Press. The protein was then fractionated using ammonium sulfate precipitation, resuspended in buffer A and, subsequently, purified on a Q-Sepharose column followed by an SP Sepharose column or a Superdex 200 column (all columns from GE Healthcare Life Sciences). N-terminal sequencing of NmClpP was carried out by sequential Edman reactions at the SPARC BioCentre at The Hospital for Sick Children, Toronto. Protein concentrations were determined by absorbance at 280 nm with extinction coefficients calculated using ProtParam (http://ca.expasy.org/tools/protparam.html) or by Bradford assay (Bio-Rad Laboratories).

**ClpP activity assays.** The activity of compound-activated ClpP was determined by fluorescence using a FITC-labeled casein substrate (Millipore-Sigma, product #C3777). Each 200 μL reaction contained 3.6 μM ClpP, 1% DMSO, 10 μM ACP1 or 1 μM ADEP, in 25 mM TrisHCl, pH 7.5, and 100 mM KCl. Reaction mixtures were pre-incubated for 10 min at 37 °C in a black 96-well plate before casein-FITC was added to a final concentration of 4.5 μM. Control reactions contained 1% DMSO, but no compound. Reactions were monitored by fluorescence (490 nm excitation, 525 nm emission) on an EnSpire plate reader (Perkin Elmer) equipped with a fluorescence module for 1 h, with a reading every 10 s. Initial rates were calculated using the initial linear slope of the degradation curve, typically within the first 10 min. All rates were normalized to the degradation rate of the EcClpP E40A mutant without compound.

**Statistics and reproducibility.** All enzymatic assays of ClpP were done at least in triplicate.

**X-ray crystal structure determination and refinement.** *E. coli* ClpP was dissolved at 10 mg/mL in 10 mM HEPES, pH 7.5, and 200 mM NaCl containing 5 mM ACP1-06 (dissolved in DMSO). The solution was mixed with and incubated above a well solution containing 51-63% 2-methyl-2,4-pentanediol (MPD) and 0.1 M sodium acetate, pH 5.0 at room temperature (RT). Diffraction data from an EcClpP+ACP1-06 crystal, flash-frozen in a boiling nitrogen stream, were collected with 0.97949 Å wavelength X-rays at the Canadian Light Source (Saskatoon, Saskatchewan, Canada), beamline 08B1-1 at −180 °C. The data were indexed and integrated using Mosflm[46], scaled using Scala[47], and merged using Xprep from Shelx[48]. The structure was solved by molecular replacement using Phaser[49] with the structure of apo-EcClpP (1YG6)[10] as a search model, refined using Phenix[50], and modeled with Coot[51]. Phenix torsion angle non-crystallographic symmetry constraints were applied to all identical chains from residues 31 to 205. All 14 copies of ACP1-06 were split into two identical copies, each atom assigned an occupancy value of 0.5, and the trifluoromethylpyridine portion rotated into the alternate density. Phenix was then used to refine the occupancy of the ACP1-06 atoms in both conformations[50]. The final refined structure has excellent geometry with 98.3% of residues in favored Ramachandran regions.

Apo-NmClpP was dissolved at 5 mg/mL in 25 mM TrisHCl, pH 7.5 and 100 mM KCl. The solution was mixed with and incubated above a well solution containing 0.2 M potassium thiocyanate and 40% MPD at RT. Diffraction data from a single NmClpP crystal, flash-frozen in a boiling nitrogen stream, were collected with 1.23984 Å wavelength X-rays at CLS beamline 08B1-1 at −180 °C. Diffraction data were indexed and integrated using Mosflm[46], scaled using Scala[47] and merged using Xprep from Shelx[48]. The structure was solved by molecular replacement using Phaser[49] with PDB id 5DKP[19] as a search model, refined using Phenix[50], and modeled with Coot[51]. Phenix torsion angle non-crystallographic symmetry constraints were applied to all identical chains from residues 23 to 199. The final refined structure has excellent geometry with 98.2% of residues in favored Ramachandran regions.

To obtain crystals of NmClpP complexed with ADEP-14, NmClpP (10 mg/mL) was incubated with 1 mM ADEP-14 for 1 h on ice, then mixed at a 1:1 ratio with well solution containing 0.2 M potassium thiocyanate and 40% MPD on sitting drop plates. The plates were incubated at 23 °C, and crystals were observed after two weeks and grew to maximum dimensions within a month. Diffraction data from flash-frozen crystals were collected at −180 °C with 1.5418 Å X-rays at the Structural Genomics Consortium-University of Toronto using a Rigaku FR-E Superbright Rotating Anode generator, then indexed, integrated, and scaled in HKL2000[52]. The structure of the NmClpP+ADEP-14 complex was determined by molecular replacement using the apo-NmClpP structure as a search model, and the solution was initially refined in Phenix with simulated annealing and coordinate shaking to remove model bias. Clear electron density for ADEP-14 was found in all 28 subunits in the asymmetric unit and the molecules were modeled at occupancy = 1.0. Subsequent modeling and refinement were performed in Coot and Phenix[50,51] using translation-libration-screw (TLS) parameters with individual coordinate, B-factor and occupancy refinement. The final refined structure has excellent geometry with 97.9% of residues in favored Ramachandran regions.

Crystals of the NmClpP E58A and NmClpP E31A+ E58A mutants were obtained by incubating 10 mg/mL of protein with well solution (1:1 ratio) containing 0.2 M potassium thiocyanate and 40% MPD using sitting drop plates. The plates were incubated at 23 °C, and crystals were observed within two weeks and allowed to grow to maximum dimensions in a month. Crystals of mutant NmClpP were harvested and diffraction data were collected as above. Diffraction data collection, processing, structure solution and refinement for both NmClpP mutants were performed as for the NmClpP+ADEP14 complex. The final refined structures have excellent geometry with >97.0% of residues in favored Ramachandran regions. Diffraction data collection and refinement statistics for all structures are summarized in Table 1.

**ClpP barrel volume measurements.** ClpP barrel volume measurements were performed in VMD 1.9.1 using the VolArea extension (Surface and Volume Calculator)[53]. All analyzed crystal structures of tetradecameric ClpP were first superimposed on that of apo-NmClpP (5DKP)[19] using the align function in PyMOL[54]. The active site chamber was defined using the analysis section specified by the following parameters: width: 52.50, height: 32.86, depth: 50.31, $x = -28.66$, $y = 31.02$, and $z = -33.61$, and a cavity probe radius of 20 Å. The analysis box for volume measurements is bounded by residue Val75 (N-terminus of the αC helix) of each subunit, located at the boundary between the axial pore cavity and the lumen of ClpP.

**Protein expression and purification for NMR.** Transformed Codon+ *E. coli* BL21 (DE3) cells were grown in minimal M9 $D_2O$ media supplemented with $^{15}NH_4Cl$ and d7-glucose as the sole nitrogen and carbon sources, respectively. Cells were grown at 37 °C and protein overexpression was induced with 0.2 mM IPTG at $OD_{600} = 1.0$. Expression was allowed to proceed overnight at 25 °C. Proteins were purified using Ni-affinity chromatography, followed by removal of the SUMO tag

using Ulp1 protease. The cleaved tag and other impurities were removed using another step of Ni-affinity chromatography. The flow-through from the second Ni-affinity purification was concentrated using an Amicon Ultra-15 50 K MWCO (Millipore) concentrator and subjected to gel filtration on a HiLoad 16/60 Superdex 200 column (GE Healthcare) containing 50 mM imidazole, 100 mM KCl, 5 mM DTT, pH 7.0. Fractions were pooled and MMTS labeling was performed as described previously[22]. Protein concentrations were determined in 8 M guanidium chloride using extinction coefficient values obtained from ExPASy's ProtParam program. Prior to NMR measurements the samples were buffer-exchanged into 100 mM KCl, 1 mM EDTA, 5 mM MgCl$_2$, and 50 mM imidazole at pH 6.7 prepared in 99.9% D$_2$O.

**NMR**. All NMR measurements were performed at 18.8 T and 40 °C using a Bruker AVANCE III HD spectrometer equipped with a cryogenically cooled pulsed-field gradient triple-resonance probe. $^1$H-$^{13}$C correlation spectra were recorded as HMQC datasets, exploiting a methyl-TROSY effect that is particularly beneficial for applications to high molecular weight proteins[55]. Spectra were processed using the NMRPipe suite of programs[56], analyzed using scripts written in-house, and visualized using Ccpnmr[57].

**SAXS experiments on NmClpP**. SAXS experiments were performed at the SAXS1 beamline at the Brazilian Synchrotron Light Laboratory (LNLS, CNPEM, Campinas, São Paulo, Brazil) using a Pilatus 300 K (Dectris) detector. Samples were collected at the concentration of 1.27 mg/mL in 50 mM TrisHCl (pH 7.5), containing 200 mM KCl, 25 mM MgCl$_2$ and 10% glycerol. ADEP-04 was used at the final concentration of 0.7 mM. All samples (in the absence or in the presence of ADEP-04) were prepared with DMSO at the final concentration of 2.8% (v/v). Data collection was performed at 20 °C, and every sample was exposed to ten frames of 10 s and one frame of 300 s. Data processing was done using the ATSAS 2.7.2 package[58]. Evaluation of X-ray damage was performed by comparing curves from different X-ray exposure times and by analyzing the linearity of their Guinier regions. Curves were scaled and merged using the DATMERGE software[59]. Molecular masses were obtained using the SAXSMoW program[60]. Pair distance distribution functions were generated using the GNOM software[61]. Restoration of solution ab initio structures was carried out using a simulated-annealing methodology implemented by the DAMMIN software and a P72 symmetry (homo-dimer of homo-heptamers) was employed[62]. Ten models were generated and compared against each other using the DAMAVER package[43]. The DAMs with the lowest normalized spatial discrepancy (NSD) values were used as reference models. Rendering of DAMs and three-dimensional high-resolution structures was performed by the UCSF Chimera software[63]. Determination of hydrodynamic properties from PDB models was done by the Hydropro software[64].

**Reporting summary**. Further information on research design is available in the Nature Research Reporting Summary linked to this article.

## Data availability

The atomic coordinates and structure factors for EcClpP + ACP1-06, Apo NmClpP, NmClpP + ADEP-14, NmClpP E58A, and NmClpP E31A + E58A, have been deposited in the Protein Data Bank under the accession codes 6NB1, 6NAQ, 6NAH, 6NAW, and 6NAY, respectively.

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

## Acknowledgements

The authors thank Aiping Dong and Wolfram Tempel of the Structural Genomics Consortium Toronto for help with the diffraction data collection. M.F.M. is supported by the Precision Medicine Initiative (PRiME) at the University of Toronto internal fellowship number PMRF2019-007. S.V. is supported by Canadian Institutes of Health Research (CIHR) postdoctoral fellowship. T.V.S. was supported by a CNPq-Brazil fellowship (202192/2015-6), a Saskatchewan Health Research Foundation postdoctoral fellowship, and currently holds a CIHR postdoctoral fellowship. V.B. is the recipient of the Ontario Graduate Scholarship (OGS) and previously held the Natural Sciences and Engineering Research Council of Canada's (NSERC) Postgraduate Scholarship-Doctoral (PGS-D) award and a Jaro Sodek Award—Ontario Student Opportunity Trust Fund (OSOTF) fellowship from the Department of Biochemistry at the University of Toronto. Y.Q.M. is supported by a fellowship from the Centre for Pharmaceutical Oncology (University of Toronto). K.R. was supported by a CIHR Training Program in Protein Folding and Interaction Dynamics: Principles and Diseases fellowship (TGF-53910) and by a University of Toronto Fellowship from the Department of Biochemistry, and currently holds an OGS. M.M.B. is supported by an OGS fellowship. J.D.G. was supported by a NSERC PGS-D2 fellowship. This work was supported by a CIHR Emerging Team Grants from the Institute of Infection and Immunity (XNE-86945) and a CIHR Project grant (PJT-148564) to R.A.B., E.F.P. and W.A.H. and by Global Affairs Canada (Canada) and CAPES (99999.004913/2015-09; Brazil) to W.A.H. and C.H.I.R. E.F.P. acknowledges support by NSERC (RGPIN-2015- 04877) and the Canada Research Chairs Program. M.B. was supported by NSERC Discovery grant (DG-20234) and CIHR new investigator program. This work was partially supported by FAPESP (2015/15822-1, 2012/01953-9, 2016/05019-0) and a CNPq to L.R.S.B. who also holds a research fellowship from CNPq (306943/2015-8, 420567/2016-0). C.H.I.R. has research fellowship from CNPq and FAPESP (2012/50161-8). The SGC is a registered charity (No. 1097737) that receives funds from AbbVie, Bayer, Boehringer Ingelheim, Genome Canada through Ontario Genomics Institute Grant OGI-055, GlaxoSmithKline, Janssen, Lilly Canada, Merck, the Novartis Research Foundation, the Ontario Ministry of Economic Development and Innovation, Pfizer, Takeda, and Wellcome Trust Grant 092809/Z/10/Z. The Canadian Light Source is supported by the Canada Foundation for Innovation, NSERC, the University of Saskatchewan, the Government of Saskatchewan, Western Economic Diversification Canada, the National Research Council Canada, and CIHR.

## Author contributions

M.F.M. and W.A.H. conceived and designed the research. M.F.M. with the help of B.T.E. and S.B. carried out all the X-ray structure determination experiments and subsequent analyses. E.F.P. helped in analyzing the X-ray data. E.L. carried out the activity assays for Fig. 4a, b and purified the proteins for structure determination. S.V. carried out the NMR experiments with the supervision of L.E.K. T.V.S. carried out the SAXS experiments with the supervision of L.R.S.B. and C.H.I.R. V.B., J.L.Z., Y.-Q.M., K.R. and M.M.B. helped with the various biochemical assays. J.D.G. synthesized the compounds under the supervision of R.A.B. S.P. carried out the sequence analysis under the supervision of M.B. M.F.M. and W.A.H. wrote the paper with the help of all the other authors.

## Competing interests

The authors declare no competing interests.
