## [Peer Review File · Communications Biology]

Reviewers' comments:

Reviewer #1 (Remarks to the Author):

Both regulatory and general proteolysis plays an important role in cellular protein homeostasis and in the control of many regulatory and developmental cellular pathways. In bacterial cells this control is mediated by AAA+ protease complexes such as E. coli ClpXP or ClpAP. Here the AAA+ unfoldases, such as ClpX or ClpA, which form hexameric rings, can recognize substrate proteins, unfold and transport them through the pore of the ring structure into the associated double heptameric ClpP protease complex. The hexameric AAA+ complex interacts via 6 peptide loops with 7 hydrophobic pockets formed by the double heptameric ClpP protease complex, where the peptidase active sites are lining the inside of the cavity formed by this barrel like structure. Importantly, small compounds such as ADEP or ACP were identified as antibiotics, which bind to these hydrophobic pockets, thereby interfering with the association of the AAA+ hexamer with the ClpP protease. At the same time the binding of these small compounds leads to an activation and dysregulation of the ClpP protease complex, which results in the recognition and uncontrolled degradation of many new cellular substrate proteins previously not recognized by the full complex. This deregulated degradation is toxic for the cells and results in the death of the bacteria. This new mechanism of antibiotic activity has to be investigated in more detail, because it represents a new mechanism of antibiotic activity and additional avenue to identify and screen for new antibiotics. In their Manuscript Leung et al use various structural biological approaches (X-ray structure, NMR and SAXS) and biochemical activity assays and different ClpP variants to investigate the molecular mechanism by which the binding interaction of two different compound classes (Acyldepsipeptides (ADEPs) and activators of compartmentalized proteases (ACPs)) result in an activation and dysregulation of the ClpP protease complex. For this they solve the structure of different ClpP's (from E.coli and Neisseria) with different ADEP variants and for the first time with an ACP compound (Fig 1, 2 &3). In addition to other changes they observe distinct structural changes with the two different compounds, which appear to cause a comparable reorganization of hydrogen bonding networks at the entrance pores (Fig 3, S2, S3). Based on these results the authors generated newly designed ClpP variants to investigate the observed conformational changes. Structural and biochemical analysis of these new ClpP variants (Fig 4, 5) together with NMR and SAXS experiments (Fig 6), also in comparison to other known ClpP structures (Fig S4, S5), support their hypotheses on the ClpP activation network they observed with the ADEP and ACP bound compound. Therefore they can present a reasonable and very interesting model of the important drug induced molecular changes resulting in the different properties of activated and dysregulated ClpP protease complex, including e.g. widened axial pores (Fig 7).

This is a well-designed whose results are really very interesting.

I just have one question:

-Is the ClpP variant ClpP(T10C) and its modified version(s) used for the NMR Experiments still functional?

Reviewer #2 (Remarks to the Author):

The manuscript by Mabanglo and colleagues contains an essentially complete report of the

conformational changes induced by various compounds in the bacterial caseinolytic protease P (ClpP), which is a cylindrical serine protease required for maintaining cellular proteostasis. Dysregulation and activation of ClpP leads to bacterial cell death. ADEPs and ACP1s are two groups of small compounds that can activate ClpP, but how they do this is not known.

The authors present a first crystal structure of an ACP1 activator with *E. coli* ClpP, a crystal structure of *N. meningitidis* ClpP with a novel ADEP analogue, and structures *N. meningitidis* ClpP mutants that mimic the structural and kinetic effects of ADEPs and ACP1s. These structures, in combination with consistent solution studies (methyl-TROSY NMR and SAXS), allowed the authors to propose that reorganization of the hydrogen bonding networks at the axial entrance pores of ClpP is necessary and sufficient for protease activation. This is a highly significant advance in understanding how ADEPs and ACP1s activate ClpP.

The manuscript is clearly written. The crystallography appears to be solid.

I have only two minor comments:

Page 8, 2nd paragraph: unexplained extra density at Ser111: please, specify that Ser111 is the catalytic nucleophile. The Figure suggests an ester with acetate, or the presence of a Glu/Gln? I guess this has been considered?

Abstract: it would be nice to mention the most important residues for the reorganization of the hydrogen bonding network.

Reviewer #3 (Remarks to the Author):

The study by Mabanglo and coworkers describes work to define the mechanism of activation of the ClpP protease by two families of small molecules.

The research question is of importance for two main reasons. First, these molecules have been shown to cause dysregulation of the protein, leading to cytotoxicity and thus they may provide therapeutic leads for bacterial infections and cancer. Second, from a fundamental scientific perspective, the activation of an enzyme by xenobiotics is less common, and generally includes more convoluted mechanisms, than enzyme inhibition and is for that reason interesting to define in terms of the detailed molecular mechanism.

I particularly appreciate the mutational studies that support the authors hypothesis regarding the changes in the intersubunit interaction pattern around the axial pores

There is a lot of work performed and some very interesting mechanistic proposals made, still there are a number of issues that I find require consideration.

Main issues.

1.

In general, the paper is well written but the reader is left with some confusion as to what is really novel and what is based on (or conflicting with) previous data.

As an example, it is stated that:

“the molecular basis of ClpP activation by ADEPs or ACP1s is still not elucidated”

While reference 15 (Lee BG, et al. Structures of ClpP in complex with acyldepsipeptide antibiotics reveal its activation mechanism. *Nat Struct Mol Biol* 17, 471-478 (2010)) discusses a number of global and local structural changes to address this question.

Also ref 39 (Stahl M, et al. Selective Activation of Human Caseinolytic Protease P (ClpP). *Angew Chem Int Ed Engl* 57, 14602-14607 (2018)) discusses structural changes for another family of inhibitors that upon first glance appear similar to the ACP1 family.

As the authors propose a consensus mechanism it would appear essential that the mechanism proposed is discussed in detail in relation to what has been previously proposed (this is not limited to the two studies above).

2.

Regarding the SAXS studies.

As several crystal structures in different conformations are available and that SAXS de novo modeling has severe limitations, I suggest that the SAXS data is first modeled using the available structures to see if they explain the data equally well (or better) than de-novo DAM models. At least these fit values should be included for comparison.

3.

As a consensus mechanism for activation/regulation is proposed, a straight-forward bioinformatic analysis regarding residue conservation in the activator binding site and the intersubunit interaction surface around the axial pores should be done. If conservation and/or pairwise covariation of the relevant residues in this region is observed it would significantly strengthen the authors hypothesis.

4.

The limited resolution of some of the structures (in particular 6NAH) calls into question the relevance of describing sidechain-sidechain interaction distances. Electron density (simulated annealing omit maps) for the region depicted in figure 3 should be provided to support the claims and allow the reader to judge the relevance of described distances.

5.

Regarding the intersubunit interaction pattern (e.g. fig 3 and S5)

Is there a reason for applying a cutoff of 4.3 Å?

This number appears arbitrary to me, it is certainly too long for a hydrogen bond.

As the interactions described include both H-bonds and charge-charge electrostatic interactions which are significantly different both in terms of requirements for distance and geometry, I suggest these are analyzed and described as separate types of interactions.

Minor issues.

1.

Regarding the 6NAH structure. Is it not possible to assign an alternate asymmetric unit to encompass two full tetradecamers rather than one tetradecamer and two half-tetradecamers?

2.

Was any map averaging used?

In principle, given the very high non-crystallographic symmetry, averaging of the maps should significantly improve the maps (basically approaching a phase error of zero if adequate cyclic averaging is applied). This could be of great value especially at the limited resolutions obtained. Of course, if there are differences between protomers, this would blur the averaged maps in the relevant regions, which may actually also provide further information in the current case. I encourage the authors to try this to see if better maps can be obtained.

RESPONSE TO REVIEWERS

Reviewer 1

Is the ClpP variant ClpP(T10C) and its modified version(s) used for the NMR Experiments still functional?

Response:

We appreciate the favorable comments of Reviewer 1 about our manuscript. We also would like to recognize that his question on the functionality of the T10C mutated-NmClpP and of the $^{13}\text{CH}_3$ -S-labeled NmClpP variant used in the NMR studies is valid. However, having the mutant active is not a pre-requisite for these NMR experiments. In these experiments, we are mapping the conformational changes of ClpP upon addition of activating compounds. Such changes are likely to occur even for an inactive ClpP mutant.

We had recently shown that the first few residues at the N-terminus (including T10 or its equivalent in other species) are critical for ClpP function, constituting a critical conformational switch whose mutation can lead to loss of function in *S. aureus* ClpP [Vahidi *et al.*, PNAS, 115(28):E6447-E6456]. Furthermore, we showed in the same study that the addition of small molecule activator such as ADEP can restore the activity of these mutants.

Thus, we expected that the T10 mutation of NmClpP and its subsequent ^{13}C -labeling, necessitated by the methyl-TROSY NMR studies for probing ClpP conformational changes, would not be exactly benign towards ClpP activity. For instance, activity assays of T10C- (unreacted), and ^{13}C -labeled (reacted) NmClpP showed observable decreases in proteolytic activities relative to wild-type NmClpP (Figure R1). The proteolytic activities measured, however, still reflect the differential activating effects of ADEP and ACPs, with ADEPs inducing greater activation of proteolytic activity in both T10C and ^{13}C -labeled NmClpP. The T10C and ^{13}C -labeled NmClpP variants have nearly identical proteolytic activities towards casein-FITC in the presence of either ADEP-14 or ACP1-17, suggesting that S-methylation of C10 at the N-terminus does not result in further decrease in proteolytic activity.

We performed a similar experiment using the fluorescent, short peptide substrate, succinyl-Leu-Tyr-aminomethylcoumarin, and found that while the T10C (unreacted NmClpP) and ^{13}C -labeled (reacted NmClpP) are still functional, their peptidase activities are reduced by about 50% relative to that of wild-type NmClpP (Figure R2).

These activity assays reflect the critical importance of N-terminal residues in ClpP function, as it encompasses the axial loops that gate the ClpP axial pore and mediates the entry of substrate proteins or peptides. However, mutation of T10 to cysteine and subsequent labeling with $^{13}\text{CH}_3$ -S was an absolute necessity to undertake methyl-TROSY NMR and measure the conformational changes at the axial pores in a facile and less-time consuming way. We felt that so long as NmClpP variants retained some activity, and as long as the general structural integrity of the ClpP barrel is sustained (as we demonstrated in ^1H - ^{13}C HMQC experiments), then the results of the methyl-TROSY NMR experiments and their interpretation are valid and reasonable. The results of Figure R1 and R2 are not included in the paper as we feel they are needed to support the NMR experiments.

Figure R1. Activity assays using wild-type (WT), reacted (R, ^{13}C -labeled), and unreacted (UR, T10C-mutant) NmClpP, using casein-FITC as substrate with ADEP-14 ($10\ \mu\text{M}$) or ACP1-17 ($100\ \mu\text{M}$) as activators. DMSO is added in control assays where no activation is expected.

Figure R2. Activity assays using wild-type (WT), reacted (R, ^{13}C -labeled), and unreacted (UR, T10C-mutant) NmClpP, using the fluorescent short peptide, succinyl-Leu-Tyr-aminomethylcoumarin as substrate, but without ADEP-14 and ACP1-17, as ClpP is in general active towards short peptides (in the absence of activators).

Reviewer 2

Comment 1.

Page 8, 2nd paragraph: unexplained extra density at Ser111: please, specify that Ser111 is the catalytic nucleophile. The Figure suggests an ester with acetate, or the presence of a Glu/Gln? I guess this has been considered?

Response:

Yes, Ser111 is the nucleophilic residue in EcClpP. We did try to model this extraneous density with many different covalent modifications including acetylation (acetate is found in the crystallization buffer), phosphorylation, and formylation. We also tested various short peptides linked to Ser111. Upon refinement, however, none of these attempts turned out to be correct with none of them modelling the extra density correctly. Sequencing of EcClpP prior to expression and crystallization confirmed that there was no mutation in the gene; therefore, Ser111 could not have been replaced with a Glu or Gln residue. Mass spectrometry analysis of the purified protein also did not yield useful results. We assume that this extra density represents a superposition of several conformations and orientations of whatever is linked covalently to Ser111 and, therefore, decided not to model the extra electron density at all.

Comment 2.

Abstract: It would be nice to mention the most important residues for the reorganization of the hydrogen bonding network.

Response:

We understand the reason for this suggestion by Reviewer 2. However, since the study aims to provide a unified mechanism for activation of ClpP across species, not just NmClpP or EcClpP, which we focused on in the present study, and that amino acid residues have different numbers in the protein sequence owing to the differences in their lengths, we feel that not specifying the residue numbers in the abstract would be more appropriate, especially given the word limitation of the abstract. The residues and their special roles, however, are discussed in detail and summarized in Figure 7a.

Reviewer 3

Comment 1.

In general, the paper is well written but the reader is left with some confusion as to what is really novel and what is based on (or conflicting with) previous data. As an example, it is stated that: “the molecular basis of ClpP activation by ADEPs or ACP1s is still not elucidated” While reference 15 (Lee BG, et al. Structures of ClpP in complex with acyldepsipeptide antibiotics reveal its activation mechanism. *Nat Struct Mol Biol* 17, 471-478 (2010)) discusses a number of global and local structural changes to address this question. Also ref 39 (Stahl M, et al. Selective Activation of Human Caseinolytic Protease P (ClpP). *Angew Chem Int Ed Engl* 57, 14602-14607 (2018)) discusses structural changes for another family of inhibitors that upon first glance appear similar to the ACP1 family. As the authors propose a consensus mechanism it would appear essential that the mechanism proposed is discussed in detail in relation to what has been previously proposed (this is not limited to the two studies above).

Response:

We thank Reviewer 3 for this suggestion. The Discussion section has been rewritten extensively to make our study’s new findings more clearly articulated in the context of existing knowledge on the mechanism of ClpP activation. This is clearly explained on page 17.

Comment 2.

Regarding the SAXS studies. As several crystal structures in different conformations are available and that SAXS de novo modeling has severe limitations, I suggest that the SAXS data is first modeled using the available structures to see if they explain the data equally well (or better) than de-novo DAM models. At least these fit values should be included for comparison.

Response:

According to Reviewer 3’s suggestion, we extracted theoretical SAXS curves from crystal structures of NmClpP, apo and ADEP-bound, and *Staphylococcus aureus* (SaClpP) in compact, compressed and extended conformations. Additionally, we calculated the theoretical SAXS curve of the SaClpP V7A mutant cryo-EM structure. This was done using the program CRY SOL. Comparison between theoretical SAXS profiles with experimental and DAM SAXS profiles (Figure R3) led to the following conclusions:

(1) at low angles, i.e. low scattering vector q values, all X-ray scattering curves (Figure R3A and R3C) behave similarly, revealing that experimental DAM and theoretical curves from bacterial ClpP possess similar radii of gyration ($\sim 43\text{-}44$ Å for NmClpP and $\sim 42\text{-}44$ Å for SaClpP) independent of conformational state;

(2) all curves in Figure R3A and R3C display mainly differences at high angles (high scattering vector q values), with none of the scattering curves from crystal structures explaining the experimental curve as good as the one from the DAMs;

(3) analyzing the decay of the SAXS curves by means of the dimensionless Kratky plots (Figure R3B and R3D), we identified differences at high qR_g -values (range from 5 to 11) that are likely due to the presence of intrinsic flexibility/dynamics of NmClpP in solution, which is not observed in the crystal/cryo-EM structures. SAXS measurements represent an averaged scattering derived from different conformers in solution and while the experimental SAXS data are from a dynamic system, theoretical scattering curves extracted from high-resolution structures represent the

conformation of a single conformer.

Therefore, based on the above observations, modeling the SAXS data using the available crystal structures does not fully explain the solution structure since it does not consider the protein dynamics. Hence, this analysis was not included in the paper.

Figure R3. Comparison of SAXS profiles from bacterial ClpP high-resolution 3D structures, DAMs and NmClpP samples in solution. **(A)** Scattering profiles and **(B)** dimensionless Kratky plot of NmClpP (apo state, experimental), NmClpP DAM (apo state) and SaClpP in different conformations. **(C)** Scattering profiles and **(D)** dimensionless Kratky plot of NmClpP (ADEP-bound state, experimental), NmClpP DAM (ADEP-bound state) and SaClpP in the same conformations as in (A) and (B).

Comment 3.

As a consensus mechanism for activation/regulation is proposed, a straight-forward bioinformatic analysis regarding residue conservation in the activator binding site and the intersubunit interaction surface around the axial pores should be done. If conservation and/or pairwise covariation of the relevant residues in this region is observed it would significantly strengthen the authors hypothesis.

Response:

We agree with Reviewer 3 that a bioinformatic analysis showing conservation of residues in ClpP sequences directly involved in activation and axial pore dynamics would enhance the proposed mechanism for activation summarized in Figure 7a. We, therefore, performed a multiple sequence alignment of 460 bacterial ClpP protein sequences (new Figure S9) from UniProt and found that the residues involved in electrostatic interactions around the axial pore are highly conserved, thus appearing as consensus residues on the Sequence Logo plot. Residues involved in axial pore ordering upon activator binding are similarly conserved. Finally, many hydrophobic residues dominate the N-terminus as consensus residues (within residues 16 to 64 in EcClpP, see Figure S7a) supporting their role in N-terminal region/axial loop organization through stabilizing hydrophobic interactions. A new Figure S9 has thus been added to the manuscript and the discussion of the ClpP activation mechanism has been modified accordingly.

Comment 4.

The limited resolution of some of the structures (in particular 6NAH) calls into question the relevance of describing sidechain-sidechain interaction distances. Electron density (simulated annealing omit maps) for the region depicted in figure 3 should be provided to support the claims and allow the reader to judge the relevance of described distances.

Response:

Simulated annealing omit maps for the regions depicted in Figure 3b-f have been added as a supplementary figure (Figure S5a-e). Changes in the body of the text have been made to reflect this. Our findings, including the distances for the electrostatic interactions, are fully supported by the simulated annealing omit maps, with the side chains of residues involved in hydrogen bonding and electrostatic interactions, fitting the omit maps reasonably well. Of note is the omit map for apo NmClpP (PDB id 6NAQ), where difference electron density for the side chain of E31(B) is definitely present, but not perfectly reflecting the shape of its carboxylate group (Figure S5c). The most abundant side chain rotamer for E31(B) was therefore modeled in the structure. While alternate rotamers for E31(B) exist, a hydrogen bonding interaction with S57(A) is possible in all cases. In fact, mutation of E31 to alanine in NmClpP was found to be activating (Figure 4a), supporting that the S57-E31 interaction is a true interaction.

Unfortunately, for Figure 3a, which shows the interface for Apo-EcClpP (PDB id 1YG6), we could not create a simulated annealing omit map since the reflection file for this structure is not available in the PDB. PDB id 1YG6 was deposited in 2005. Nevertheless, the resolution for the apo-EcClpP structure is sufficiently high at 1.85 Å with a bond r.m.s.d. of 0.011 Å, indicating that the distance measurements are very reliable.

Comment 5.

Regarding the intersubunit interaction pattern (e.g. fig 3 and S5) - Is there a reason for applying a cutoff of 4.3 Å? This number appears arbitrary to me, it is certainly too long for a hydrogen bond. As the interactions described include both H-bonds and charge-charge electrostatic interactions which are significantly different both in terms of requirements for distance and geometry, I suggest these are analyzed and described as separate types of interactions.

Response:

The cut-off distance (4.3 Å) in Figure 3a was chosen because we decided to use the interface between Chains D and E to describe the electrostatic interactions around the EcClpP axial pores, and to be consistent in our use of chain pairs (i.e. Figure 3a-c use Chains D and E). Apo-EcClpP (PDB id 1YG6) is a tetradecamer with a total of fourteen E67-R36 interaction pairs, with distances ranging from 3.2 Å (E67 of Chain M, R36 of Chain N) to 8.7 Å (E67 of Chain E, R36 of Chain F). The distance depends on the captured rotamers of the pairing residues, both Glu (E) and Arg (R) residues are branched amino acids that can assume a variety of rotameric states. In apo NmClpP (PDB id 6NAQ), the equivalent interaction distance (E58-R27) ranges from 2.6 to 3.9 Å in the crystal structure, i.e. *all* of the E58-R27 pairs assume rotamers better positioned for closer interaction.

Since the assignment of Chain ID's in ClpP is arbitrary, in retrospect, we could have chosen any one of the 14 chain pairs for Figure 3a, especially Chains M and N where the interaction distance is shortest at 3.2 Å. However, a 4.3 Å distance is reasonable for an ionic bond, which is how we described the E67-R36 interaction in the manuscript. On the contrary, the second interaction present in NmClpP but not EcClpP (S57-E31, Figure 3e) is a hydrogen bond (distance 3.3 Å, Figure 3d). A hydrogen bond, like an ionic bond, is an electrostatic interaction. In the manuscript, we used the term 'electrostatic' as a more general term that encompasses both ionic and hydrogen bonds.

We would like to note that the interaction distances depend on the rotamers trapped in the crystal. A longer E67-R36 interaction distance such as that found between Chains E and F (8.7 Å) of EcClpP corresponds to weaker electrostatic interaction. We argue that, despite the rotamer-dependent range of interaction distance for this pair of residues, the activating mutation Glu to Ala in both EcClpP and NmClpP (Figure 4a) strongly provides evidence of the importance of abolishing this interaction to achieve activation. In this regard, whereas PDB id 6NAQ is a structure that amply represents the apo-state of NmClpP, we can surmise that a similar apo-state for EcClpP must exist, where the E67-R36 distances fall within a narrower range than is found in PDB 1YG6. We therefore chose to keep Figure 3a as it is.

Minor issues Comment 1.

Regarding the 6NAH structure. Is it not possible to assign an alternate asymmetric unit to encompass two full tetradecamers rather than one tetradecamer and two half-tetradecamers?

Response:

We thank Reviewer 3 for this helpful suggestion. Certain space groups have equivalent asymmetric units. An alternate asymmetric unit for 6NAH encompassing two full tetradecamers exists in space

group P2₁. The choice of the asymmetric unit originally used in Figure S1d was made to demonstrate how molecule packing in the NmClpP+ADEP-14 crystal prevented axial loop ordering even in the presence of an activator. Figure S1d has thus been modified to achieve the same objective while highlighting an alternate asymmetric unit (shown in green) that is more biologically relevant and informative to the general reader.

Minor issues Comment 2.

Was any map averaging used? In principle, given the very high non-crystallographic symmetry, averaging of the maps should significantly improve the maps (basically approaching a phase error of zero if adequate cyclic averaging is applied). This could be of great value especially at the limited resolutions obtained. Of course, if there are differences between protomers, this would blur the averaged maps in the relevant regions, which may actually also provide further information in the current case. I encourage the authors to try this to see if better maps can be obtained.

Response:

Yes, non-crystallographic symmetry averaging was used during the refinement of structures for EcClpP+ACP1-06 (PDB id 6NB1), apo NmClpP (PDB id 6NAQ), and the two NmClpP mutants (PDB id 6NAW and 6NAY). For NmClpP+ADEP-14 (PDB id 6NAH), we did not perform NCS averaging during refinement, and we thank Reviewer 3 for pointing out this oversight, especially given the lower resolution for this structure (2.7 Å) relative to the others in this study. We, therefore, performed another round of refinement for NmClpP+ADEP14 with NCS averaging, for 5 cycles. This resulted in a $R_{\text{work}}/R_{\text{free}}$ of 19.2/25.5, compared to a $R_{\text{work}}/R_{\text{free}}$ of 19.6/25.1 when refined without NCS averaging. Comparison of electron density maps in regions of interest discussed in the paper (axial pore interacting residues, hydrophobic binding site of ADEP14) did not show any significant difference in map quality (see Figure R4 below). The same is true for electron density maps for the rest of the NmClpP+ADEP-14 structure. Even in regions where individual subunits might show distinct rotamers such as near the axial pores (S57, E58, R27 and E31, shown in the Figure R4 below), refining with or without NCS averaging did not significantly improve the electron density maps, and the final $R_{\text{work}}/R_{\text{free}}$ values did not differ significantly. Hence, there are no further changes made in the manuscript in this regard.

Figure R4. Comparison of $2F_o - F_c$ electron density maps contoured at 0.9σ for NmClpP+ADEP14, focusing on the hydrophobic binding site where ADEP-14 binds (top panels) and the active site (bottom panels). Two adjacent subunits are coloured green and purple, while ADEP-14 is coloured orange. Residues that form electrostatic interactions (E58, S57, R27, and E31) and the catalytic triad (S102, H127, D178) are labeled. Refinement with or without non-crystallographic symmetry averaging did not significantly affect electron density map quality despite the highly symmetrical ClpP structure.

REVIEWERS' COMMENTS:

Reviewer #2 (Remarks to the Author):

The authors have addressed my questions and comments to my full satisfaction.

Reviewer #3 (Remarks to the Author):

I thank the authors for their thorough consideration of the comments.

All my comments regarding this paper has been satisfactorily resolved and the paper improved. I recommend publication of the revised manuscript.